# FEDERATED MAXIMUM LIKELIHOOD INVERSE REINFORCEMENT LEARNING WITH CONVERGENCE GUARANTEE

## ABSTRACT

Inverse Reinforcement Learning (IRL) aims to recover the latent reward function and corresponding optimal policy from observed demonstrations. Existing IRL research predominantly focuses on a centralized learning approach, not suitable for real-world problems with distributed data and privacy restrictions. To this end, this paper proposes a novel algorithm for federated maximum-likelihood IRL (F-ML-IRL) and provides a rigorous analysis of its convergence and time-complexity. The proposed F-ML-IRL leverages a dual-aggregation to update the shared global model and performs bi-level local updates – an upper-level learning task to optimize the parameterized reward function by maximizing the discounted likelihood of observing expert trajectories under the current policy and a low-level learning task to find the optimal policy concerning the entropy-regularized discounted cumulative reward under the current reward function. We analyze the convergence and time-complexity of the proposed F-ML-IRL algorithm and show that the global model in F-ML-IRL converges to a stationary point for both the reward and policy parameters within finite time, i.e., the log-distance between the recovered policy and the optimal policy, as well as the gradient of the likelihood objective, converge to zero. Finally, evaluating our F-ML-IRL algorithm on high-dimensional robotic control tasks in MuJoCo, we show that it ensures convergences of the recovered reward in decentralized learning and even outperforms centralized baselines due to its ability to utilize distributed data.

## 1 INTRODUCTION

Inverse learning is the problem of modeling the preferences and goals of an agent using its observed behavior Arora & Doshi (2020). When the behavior of a human expert is observed through demonstration trajectories containing state and action data, Inverse Reinforcement Learning (IRL) models the policy through a Markov Decision Process (MDP) to recover the latent reward function and potentially replicate the human expert's optimal policy Russell (1998). The learned reward function can support various downstream tasks such as agent modeling and transfer learning Sutton & Barto (2018); Arora & Doshi (2020). Recent work has developed provably-efficient IRL algorithms, such as Generative Adversarial Inverse Learning (GAIL) Ho & Ermon (2016) and Maximum-likelihood IRL (ML-IRL)Ratia et al. (2012); Zeng et al. (2022), all using a centralized learning approach. However, demonstration data in practice are often distributed across decentralized clients, e.g., devices, cars, and households. It is not realistic to assume that such sensitive data can always be shared or collected for centralized inverse learning, due to privacy restrictions.

To enable collaborative training of machine learning models among decentralized clients under the privacy restrictions, FL provides a promising solution by maintaining training data on local devices and aggregating local updates to build a global model. However, most existing work on FL consider only the forward learning problem, e.g., loss minimization Li et al. (2019), policy improvement Jin et al. (2022), learning with heterogeneous models Zhou et al. (2024), and efficient optimization methods Li et al. (2020; 2021b;a); Wang et al. (2020), and not the IRL problem. We note that IRL using decentralized clients and distributed data is an open problem. It often has a bi-level structure of maximizing the probability of observing expert trajectories under the current policy and optimizing discounted cumulative reward for the target reward function, which must be solved jointly during IRL. A naive integration of FL and IRL may not achieve convergence. A decentralized learning framework for IRL with theoretical analysis of the convergence and the time-complexity remains a significant challenge.

The goal of this paper is to develop a novel framework for federated maximum-likelihood IRL (F-ML-IRL) and provide a rigorous convergence/time-complexity analysis of the proposed algorithms. We adopt the Maximum Likelihood IRL (ML-IRL) approach in Zeng et al. (2022) and consider the problem of decentralized IRL of a shared latent reward function, from distributed data and using decentralized client devices. Our solution attains the privacy-preserving benefits of FL in IRL. To address the bi-level nature of IRL, our proposed algorithm's local training round McMahan et al. (2017) encompasses an upper-level learning task (on each client with local dataset) to optimize the parameterized reward function to maximize the discounted likelihood of observing expert trajectories under the current policy, as well as a low-level learning task to find the optimal policy concerning the entropy-regularized discounted cumulative reward for the current reward function. Then, we design a dual-aggregation method for aggregating both the action-value networks and reward function models every $T$ local rounds, rather than just one set of model parameters in standard FL. Further, we leverage Soft Q-learning Haarnoja et al. (2017) as the base RL algorithm. Instead of fully solving the forward RL problem before updating the reward parameter, we perform one-step updates for both the recovered policy and reward parameter alternately to improve the efficiency. To the best of our knowledge, this is the first proposal to formulate and solve this F-ML-IRL problem.

We conduct a rigorous convergence/time-complexity analysis of the proposed F-ML-IRL algorithm. Due to the tight coupling between the reward parameters and the recovered policy in IRL's bi-level optimization, the dual-aggregation method in our F-ML-IRL must be analyzed to understand its impact on convergence. By bounding the logarithmic distance between the estimated policy and the optimal policy by the distance between their corresponding Q-values, we control the variance introduced by local training by considering the time immediately after each global aggregation. Utilizing the $\gamma$-contraction property of soft Q-values, we establish the contraction property of the targeted distance, which allows us to provide a convergence proof for the policy estimate. Moreover, we leverage the Lipschitz continuity of the reward parameter and the convergence of the policy estimate to show that the gradient of the global reward parameter converges to zero as the number of communications increases. These techniques enable us to show that F-ML-IRL's policy estimation and reward optimization both converge in finite time. The change in convergence speed due to the use of only decentralized clients and distributed data (rather than centralized learning) is characterized.

Our F-ML-IRL is implemented and evaluated on high-dimensional robotic control tasks in MuJoCo Todorov et al. (2012). We compared its performance with several centralized learning baseline including Behavior Cloning (BC) Pomerleau (1988), Generative Adversarial Imitation Learning (GAIL) Ho & Ermon (2016), and IRL methods like f-IRL Ni et al. (2021) and ML-IRL Zeng et al. (2022). We consider non-iid data distribution, where clients have different local demonstration data with varying performance levels. The baselines are evaluated using centralized data with two setups (i) a single client with medium-level demonstrations and (ii) a single client with a mixture of demonstrations of different levels. The results show that our F-ML-IRL could effectively leverage distributed data and client devices in learning, to achieve similar or better recovered reward than the baselines, while meeting decentralization and data privacy restrictions. Our evaluation code is available at https://anonymous.4open.science/r/F-ML-IRL/. The key contributions of this paper are summarized as follows:

- We propose a novel framework for federated maximum-likelihood IRL (F-ML-IRL). It enables decentralized IRL of a shared latent reward function, from distributed data and using decentralized client devices, under data privacy restrictions.

- To support the bi-level optimization structure in IRL – for jointly updating the optimal policy and the reward function estimate, the proposed F-ML-IRL algorithm leverages a dual-aggregation of the model parameters, which ensures convergence to optimal results.

- The convergence and time-complexity of the proposed F-ML-IRL algorithm is quantified, with respect to local rounds $T$ and aggregation steps $M$. We show that F-ML-IRL achieves convergence in finite time and will have faster convergence with a smaller local rounds $T$.

- Our solution is evaluated on high-dimensional robotic control tasks in MuJoCo and is shown to achieve similar or higher recovered reward than a number of Imitation Learning and IRL baselines that employ centralized learning.

## 2 RELATED WORK AND BACKGROUND

IRL aims to learn the reward function using expert demonstration data, which frees the forward RL problem from the requirement of specifying the reward function beforehand Ng et al. (2000) and also facilitates imitation learning by using the recovered reward function to derive an effective policy

Abbeel & Ng (2004). Various formulations and solutions for the IRL problem have been explored. The Maximum Margin Planning algorithm frames the problem within a quadratic programming context Ratliff et al. (2006). Bayesian IRL models infer the posterior distribution of the reward function given a prior Ramachandran & Amir (2007). Probabilistic maximum entropy IRL methods favor stochastic policies using entropy regularization. In recent years, Generative Adversarial Imitation Learning (GAIL) Ho & Ermon (2016) has adopted a Generative Adversarial Networks (GANs) Goodfellow et al. (2020) framework to recover the expert's policy. In this framework, a generator proposes new policies to confuse the discriminator, while the discriminator determines whether the state-action pair from the generator's policy originates from the expert. However, existing work has not considered the IRL problem with distributed data and decentralized clients, under data privacy.

ML-IRL models the policy through a MDP and recover the latent reward function based on maximum likelihood principle. The convergence of centralized ML-IRL with a single client has been analyzed Ratia et al. (2012); Zeng et al. (2022) and is shown to outperform other IRL methods. ML-IRL considers a MDP defined by the tuple $(\mathcal{S}, \mathcal{A}, \mathcal{P}, \eta, r, \gamma)$, where $\mathcal{S}$ and $\mathcal{A}$ represent the state space and action space, respectively. $\mathcal{P}(s'|s, a)$ denotes the state transition probability, $\eta(\cdot)$ is the initial state distribution, $r(s, a)$ is the reward function, and $\gamma$ is the discount factor. Let $\theta$ denote the parameter vector for the reward function, making the reward function $r(s, a; \theta)$. The IRL problem states that the expert's behavior is characterized by a stochastic policy $\pi_{r_\theta}(\cdot|s)$. The dataset $\mathcal{D} := \{\tau_m\}_{m=1}^K$ contains trajectories $\tau_m = \{(s_t, a_t)\}_{t=0}^\infty$ from the expert policy $\pi_{r_\theta}(\cdot|s)$.

The discounted log-likelihood of observing all sample trajectories $\mathcal{D}$ from the expert is given by:

$$\mathbb{E}_{\tau \sim \mathcal{D}} \left[ \sum_{t \geq 0} \gamma^t \left( \log \pi_{r_\theta}(a_t|s_t) + \log \mathcal{P}(s_{t+1} \mid s_t, a_t) \right) \right]. \tag{1}$$

Assume the state transition probabilities $\mathcal{P}(s_{t+1}|s_t, a_t)$ are known. Then, maximizing the discounted log-likelihood is equivalent to maximizing equation 2.

$$l(\theta) = \mathbb{E}_{\tau \sim \mathcal{D}} \left[ \sum_{t \geq 0} \gamma^t \log \pi_{r_\theta}(a_t|s_t) \right]. \tag{2}$$

ML-IRL aims to maximizing $l(\theta)$ under the constraint that $\pi_{r_\theta}$ is the optimal policy targeting the discounted cumulative reward regularized by the entropy of the policy, i.e. $\pi_{r_\theta} := \arg\max_\pi \mathbb{E}_\pi[\sum_{t=0}^\infty \gamma^t(r(s_t, a_t; \theta) + \mathcal{H}(\pi(\cdot \mid s_t)))]$, where the entropy of the policy is defined as $\mathcal{H}(\pi(\cdot|s)) := -\sum_{a \in \mathcal{A}} \pi(a|s) \log \pi(a|s)$. Incorporating the policy entropy term as a regularization makes the IRL problem well-defined. This adjustment encourages the agent to explore all possible trajectories in the environment, leading to a more stochastic policy with better generalization capabilities.

For decentralized learning, FL focuses on scenarios where multiple clients work together to train a model using distributed data. FL considers the objective of the form:

$$\min_{w \in \mathbb{R}^d} f(w) \quad \text{where} \quad f(w) = \frac{1}{n} \sum_{i=1}^n f_i(w) \tag{3}$$

We assume there are $n$ clients over which the local data $D_i$ is stored. Prior to federated averaging (FedAvg), most works in FL based on Stochastic Gradient Descent (FedSGD) Shokri & Shmatikov (2015) ignored the impact of data heterogeneity and imbalance. FedAvg derives from FedSGD but allows multiple rounds of local update $\omega^i \leftarrow \omega^i - \alpha \nabla f_i(\omega^i)$ by gradient descent before aggregating the model parameters at the central server, reducing the frequency and cost of communications. The convergence of FedAvg on non-i.i.d. data has been proved Li et al. (2019). Since Fed-Avg was proposed as the vanilla FL algorithm, efficient federated optimization methods like FedProx Li et al. (2020) FedBN Li et al. (2021b), MOON Li et al. (2021a), and FedNova Wang et al. (2020) have been developed to address non-i.i.d. data and accelerate the model training process Konečný et al. (2016). Additionally, the convergence of model-heterogeneous FL, where reduced-size models are extracted from the global model and applied to low-end clients, was provided in Zhou et al. (2024). However, existing FL methods could not be directly applied to the ML-IRL problem with decentralized clients, since ML-IRL requires a bi-level optimization involving both policy improvement and reward estimate using maximum likelihood. New algorithms needs to be developed for decentralized ML-IRL with rigorous convergence/time-complexity analysis.

# 3 Federated Maximum-Likelihood IRL

## 3.1 Our Problem Statement

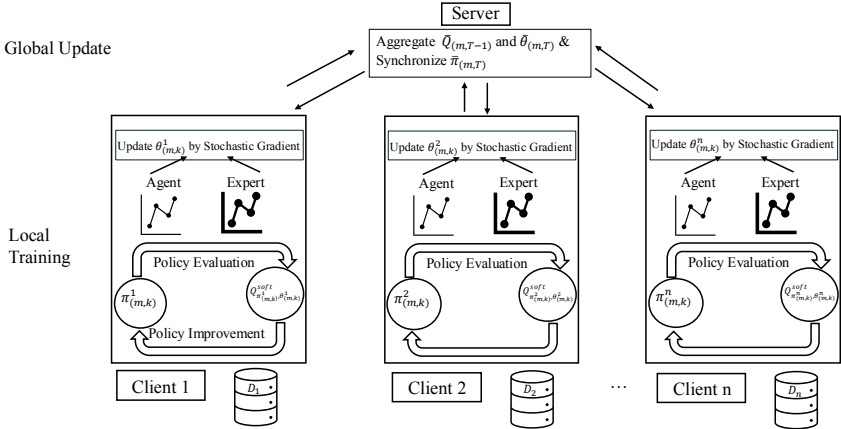

Figure 1: Our F-ML-IRL problem. It aims to recover reward function $r_\theta$ from sensitive data/demonstrations $\mathcal{D}_1, \ldots, \mathcal{D}_n$ that are distributed over $n$ clients. This requires a novel decentralized algorithm to solve a bi-level optimization – optimizing the parameterized reward function with maximum likelihood and optimizing the corresponding policy concerning the entropy-regularized discounted cumulative reward. We prove the convergence and the time-complexity of F-ML-IRL.

We consider a decentralized inverse learning problem to recover a common reward function $r_\theta$ from distributed datasets spread across $n$ clients. Due to privacy requirements, the clients cannot directly share their data for learning. Specifically, we consider $n$ clients, each with a dataset $\mathcal{D}_i := \{\tau_m^i\}_{m=1}^K$ containing trajectories $\tau_m^i = \{(s_t, a_t)\}_{t=0}^\infty$ from the $i$-th expert policy $\pi_{r_\theta}^i(\cdot|s)$. Different from centralized learning, the clients each have their local model trained on local data. By modeling the distributed expert trajectories as an MDP $(\mathcal{S}, \mathcal{A}, \mathcal{P}, \eta, \gamma)$, our goal is to learn a common reward function $r_\theta$ – parameterized by $\theta$ – from distributed data and to recover the corresponding optimal policy $\pi_{r_\theta}$. The F-ML-IRL in this paper is formulated as follows:

$$\max_{\theta \in \mathbb{R}^d} \quad L(\theta) = \frac{1}{n} \sum_{i=1}^n l_i(\theta)$$

$$\text{s.t.} \quad \pi_{r_\theta} = \arg\max_\pi \mathbb{E}_\pi \left[ \sum_{t=0}^\infty \gamma^t \left( r_\theta(s_t, a_t) + \mathcal{H}(\pi(\cdot \mid s_t)) \right) \right] \tag{4}$$

$$\text{where} \quad l_i(\theta) = \mathbb{E}_{\tau \sim \mathcal{D}_i} \left[ \sum_{t \geq 0} \gamma^t \log \pi_{r_\theta}(a_t|s_t) \right]$$

where $l_i(\theta)$ is the local likelihood calculated using client $i$'s local data $\mathcal{D}_i$ and target policy $\pi_{r_\theta}$, which further depends on the current reward function $r_\theta$ that is shared by all clients, making it a difficult bi-level optimization. We cannot directly apply FL to this problem, because the maximum likelihood problem on $L(\theta)$ depends on the recovered policy $\pi_{r_\theta}$, while the policy search for an optimal $\pi_{r_\theta}$ futher relies on the estimation of the reward function parameter $\theta$. Thus, the two-level optimization are entangled with each other and requires a new aggregation strategy in F-ML-IRL.

## 3.2 Our Proposed F-ML-IRL Algorithm

We present F-ML-IRL algorithm to solve the decentralized inverse learning problem. Our proposed solution includes three modules - local policy improvement, local reward optimization, and global bi-level aggregation. Each round of F-ML-IRL algorithm consists of $T$ local client steps running in parallel and a global server aggregation of selected model parameters at the end of each round. At each local step, each client $i$ first executes (in parallel) a policy update (on local data $D_i$) through policy evaluation and improvement steps based on soft-Q learning to address the lower-level problem. Second, each client carries out a reward optimization, where the reward parameter gradient update is derived by contrasting sampled trajectories from both the expert policy and the current

policy estimate. Next, after every $T$ local steps and at the end of round $m$, we perform a dual aggregation of both the action-value function and the reward parameters, i.e., to synchronize the local bi-level optimization of both policy and reward on decentralized clients.. While our solution is inspired by FL, F-ML-IRL performs a dual aggregation with respect to the bi-level optimization in ML-IRL. The algorithm details are presented below. Its convergence and time-complexity are rigorously analyzed in this paper.

Our F-ML-IRL is illustrated in Fig. 1. Different expert demonstration data $D_i$ are stored at different client devices. We perform local training for policy evaluation and improvement based on soft Q-learning to improve the local policy $\pi^i_{(m,k)}$ under current reward parameter $\theta^i_{(m,k)}$. We then sample trajectories from the current local policy and the expert demonstration data $D_i$, to provide an update for the reward parameter $\theta^i_{(m,k)}$. At local step $k$ of round $m$, we use $Q^{\text{soft}}_{\pi^i_{(m,k)},\theta^i_{(m,k)}}(s,a)$ to denote the action-value function (i.e., Q-value) for action $a$ and state $s$, with respect to the current policy estimation $\pi^i_{(m,k)}$ under current reward parameter estimation $\theta^i_{(m,k)}$, on each client $i$. After every $T$ steps of local training, we perform dual aggregation of the Q-values $\overline{Q}^{\text{soft}}_{(m,T-1)}$ and the reward parameter $\overline{\pi}_{(m,T)}$. To the best of our knowledge, this is the first paper considering an ML-IRL problem in this FL context.

**Local training for policy improvement.** Iterations of local training on each local client start with a shared model with parameters $\pi^i_{(m,0)}(\cdot|s)$ and $\theta^i_{(m,0)}$. During each local training round, we first evaluate the local policy $\pi^i_{(m,k)}(\cdot|s)$ by computing the Q-values $Q^i_{(m,k)}(\cdot,\cdot)$ under the fixed reward parameter $\theta^i$ for the $i$-th local client using the definitions of the soft value and soft Q functions in equation 5.

$$V^{\text{soft}}_{\pi^i_{(m,k)},\theta^i_{(m,k)}}(s) = \mathbb{E}_{\pi^i_{(m,k)},s_0=s} \sum_{t=0}^{\infty} \gamma^t \left( r(s_t,a_t;\theta^i_{(m,k)}) + \mathcal{H}(\pi_{\theta^i_{(m,k+1)}}(\cdot|s_t)) \right)$$

$$Q^{\text{soft}}_{\pi^i_{(m,k)},\theta^i_{(m,k)}}(s,a) = r(s,a;\theta^i_{(m,k)}) + \gamma \mathbb{E}_{s'\sim\mathcal{P}(\cdot|s,a)} V^{\text{soft}}_{\pi^i_{(m,k)},\theta^i_{(m,k)}}(s') \tag{5}$$

Then, $\pi^i_{(m,k+1)}(\cdot|s)$ is updated according to the policy improvement step using soft Q-learning in equation 6. It does not assume an explicit policy function but uses the Boltzmann distribution of the Q function, making the probability of choosing an action at some state $s$ proportional to the exponential of the Q-value of this action-state pair.

$$\pi^i_{(m,k+1)}(a|s) \propto \exp(Q^{\text{soft}}_{\pi^i_{(m,k)},\theta^i_{(m,k)}}(s,a)), \forall s \tag{6}$$

**Local training for reward optimization.** For the optimization towards the local reward parameter $\theta^i_{(m,k+1)}$, a stochastic gradient ascent method is proposed. The gradient of each local likelihood function $l_i(\theta)$ is given by equation 7, which derives from Lemma 1 in Zeng et al. (2022).

$$\nabla l_i(\theta) = \mathbb{E}_{\tau_i\sim\mathcal{D}_i} \sum_{t\geq 0} \gamma^t \nabla_\theta r(s_t,a_t;\theta) - \mathbb{E}_{\tau_i\sim\pi_\theta} \sum_{t\geq 0} \gamma^t \nabla_\theta r(s_t,a_t;\theta). \tag{7}$$

We construct a stochastic estimator of the exact gradient $\nabla l_i(\theta^i_{(m,k)})$, approximating the optimal policy $\pi_{r_{\theta^i_{(m,k)}}}$ with the current policy $\pi^i_{(m,k+1)}$. Specifically, we sample one expert trajectory $\tau^{E_i}_{(m,k)} := \{s_t,a_t\}_{t\geq 0}$ from the local dataset $\mathcal{D}_i$ and one agent trajectory $\tau^{A_i}_{(m,k)} := \{s_t,a_t\}_{t\geq 0}$ from the current policy $\pi^i_{(m,k+1)}$. Then we use a stochastic estimate $g^i_{(m,k)}$ to approximate the exact gradient of the local likelihood objective function $l_i$ for each local client in equation 8. The update of the reward relies on both the local softmax policy $\pi^i_{(m,k+1)}$ through $\tau^{A_i}_{(m,k)}$ and the local data $\mathcal{D}_i$ through $\tau^{E_i}_{(m,k)}$.

$$g^i_{(m,k)} = h(\theta^i_{(m,k)};\tau^{E_i}_{(m,k)}) - h(\theta_{(m,k)};\tau^{A_i}_{(m,k)}) \tag{8}$$

where $h(\theta;\tau) = \sum_{t\geq 0} \gamma^t \nabla_\theta r(s_t,a_t;\theta)$. Finally, the local reward parameter $\theta^i_{(m,k)}$ is updated as:

$$\theta^i_{(m,k+1)} = \theta^i_{(m,k)} + \alpha g^i_{(m,k)} \tag{9}$$

where $\alpha$ is the learning rate of the reward parameter update.

**Bi-level model aggregation.** Every $T$ local iterations, local Q-values and local reward parameters are communicated to the global server for aggregation, while the policy synchronization is performed based on the aggregated Q-values such that each local client has the same policy after the aggregation. We design the dual aggregation step after thorough thoughts. The reward update in equation 9 depends on how well the trajectories from policy $\pi^i_{(m,k)}$ approximates the optimal policy $\pi_{r_{\theta^i_{(m,k)}}}$, while the policy $\pi^i_{(m,k)}$ relies on the Q-value update from equation 5. Therefore, our FL algorithm aims to improve the Q-value estimates for local clients by aggregating their Q-values equation 10.

$$\overline{Q}^{\text{soft}}_{(m,T-1)}(\cdot,\cdot) := \sum_{j=1}^{N} Q^j_{(m,T-2)}(\cdot,\cdot)/N \tag{10}$$

We note that when the Q-values are represented by another network with parameter $\psi$, the aggregation of the Q-values will simply become aggregation of model parameters. The policy synchronization is automatically performed by policy improvement based on the aggregated Q-values and sent to each local client for update such that each local client has the same policy after the Q aggregation inequation 11:

$$\overline{\pi}_{(m,T)}(\cdot|s) \propto \exp(\overline{Q}^{\text{soft}}_{(m,T-1)}(s,\cdot)), \forall s \in S \tag{11}$$

Since ML-IRL requires a bi-level problem with respect to both the reward parameter and the recovered policy, we consider a dual aggregation that also applies to the reward parameter $\theta$:

$$\overline{\theta}_{(m,T)} := \sum_{j=1}^{N} \theta^j_{(m,T-1)}/N \tag{12}$$

After each dual aggregation, the global policy and reward parameters are sent to each local clients as an initialization for future local training: $\pi^i_{(m,0)}(\cdot|s) = \overline{\pi}_{(m-1,T)}(\cdot|s)$ and $\theta^i_{(m,0)} = \overline{\theta}_{(m-1,T)}$ for all $i = 1, 2, \dots, N$. The entire process of the F-ML-IRL algorithm is summarized in Algorithm 1.

---

**Algorithm 1** Federated Maximum Likelihood Inverse Reinforcement Learning (F-ML-IRL)

---
1: **Input**: Initialize reward parameter $\theta^i_{(0,0)}$ and policy $\pi^i_{(0,0)}$. Set the aggregation period to be $T$, number of local server to be $N$, and reward parameter's local stepsize as $\alpha$.
2: **for** $m = 0, 1, \dots, M-1$ **do**
3:      **if** $m > 0$ **then**
         Inherit $\pi^i_{(m,0)}(\cdot|s)$ and $\theta^i_{(m,0)}$ from last aggregation
4:      **end if**
5:      **for** $k = 0, \dots, T-2$ **do**
6:          **for** $i = 1, 2, \dots, N$ **do**
7:              Compute $Q^{\text{soft}}_{r_{\theta^i_{(m,k)}}, \pi^i_{(m,k)}}(\cdot,\cdot)$ using equation 5
8:              Update $\pi^i_{(m,k+1)}(\cdot|s)$ based on equation 6
9:              Sample an expert trajectory $\tau^{E_i}_{(m,k)}$ from local dataset $D_i$
10:            Sample a trajectory $\tau^{A_i}_{(m,k)}$ from current policy $\pi^i_{(m,k+1)}$
11:            Estimating gradient $g^i_{(m,k)}$ following equation 8
12:            Update reward parameter $\theta^i_{(m,k+1)}$ using equation 9
13:          **end for**
14:      **end for**
15:      Set $k = T-1$
16:      **for** $i = 1, 2, \dots, N$ **do**
17:          Aggregate $\overline{Q}^{\text{soft}}_{(m,k)}(\cdot,\cdot)$ by equation 10
18:          Synchronize policies $\overline{\pi}_{(m,k+1)}(\cdot|s)$ using equation 11
19:          Aggregate reward parameters $\overline{\theta}_{(m,k+1)}$ by equation 12
20:      **end for**
21: **end for**

---

# 4 THEORETICAL ANALYSIS

## 4.1 ASSUMPTIONS

**Ergodicity.** For any policy $\pi$, assume the Markov chain with transition kernel $\mathcal{P}$ is irreducible and aperiodic under policy $\pi$. Then there exist constants $\kappa > 0$ and $\rho \in (0, 1)$ such that

$$\sup_{s \in S} \|\mathbb{P}(s_t \in \cdot \mid s_0 = s, \pi) - \mu_\pi(\cdot)\|_{TV} \le \kappa \rho^t, \quad \forall t \ge 0 \tag{13}$$

where $\| \cdot \|_{TV}$ is the total variation (TV) norm, and $\mu_\pi$ is the stationary state distribution under $\pi$.

Equation 13 states that the Markov chain mixes at a geometric rate. This is a common assumption in the RL literature, which holds for any time-homogeneous Markov chain with a finite state space.

**Bounded Gradient and Lipschitz Property**. For any $s \in \mathcal{S}$, $a \in \mathcal{A}$, and any reward parameter $\theta$, the following conditions hold, where $L_r$ and $L_g$ are positive constants:

$$\|\nabla_\theta r(s, a; \theta)\| \le L_r, \ \text{ and } \ \|\nabla_\theta r(s, a; \theta_1) - \nabla_\theta r(s, a; \theta_2)\| \le L_g \|\theta_1 - \theta_2\|, \tag{14}$$

Equation 14 posits that the parameterized reward function has a bounded gradient and is Lipschitz smooth, which is common in the literature.

## 4.2 IMPORTANT LEMMAS

We first introduce two important lemmas that are used repeatedly in the converge analysis. Due to space limitations, the proofs of these lemmas are included in the appendix.

**Lemma 1.** *Suppose the above assumptions hold. Given any reward parameters $\theta_1$ and $\theta_2$, the following results hold for any $s \in \mathcal{S}$ and $a \in \mathcal{A}$:*

$$\left| Q^{soft}_{r_{\theta_1}, \pi_{\theta_1}}(s, a) - Q^{soft}_{r_{\theta_2}, \pi_{\theta_2}}(s, a) \right| \le L_q \|\theta_1 - \theta_2\|, \tag{15}$$

$$\|\nabla L(\theta_1) - \nabla L(\theta_2)\| \le L_c \|\theta_1 - \theta_2\|, \tag{16}$$

*where $Q^{soft}_{r_\theta, \pi_\theta}(\cdot, \cdot)$ denotes the soft Q-function under the reward function $r(\cdot, \cdot; \theta)$ and the policy $\pi_\theta$.*

Lemma 1 is directly derived from Lemma 2 in Zeng et al. (2022), where the positive constants $L_q$ and $L_c$ are also defined. The Lipschitz properties of the Q-value function and the gradient of the log-likelihood are essential for convergence analysis, as they help control the distance between local and global models in the FL setting.

**Lemma 2.** *For any two policies $\pi(a|s)$ and $\pi'(a|s)$, the difference in their soft Q-values under some reward function $r$ for a given state-action pair $(s, a)$ is bounded as follows:*

$$\|Q^{soft}_{r_\theta, \pi} - Q^{soft}_{r_\theta, \pi'}\|_\infty \le \frac{\gamma}{1 - \gamma} \|\log(\pi) - \log(\pi')\|_\infty \tag{17}$$

Controlling the distance between soft Q-values under different policies helps us analyze the optimality of the global policy with respect to the global reward parameter after aggregations.

## 4.3 MAIN CONVERGENCE RESULT

**Theorem 1.** *Under the above two assumptions, if we choose step size $\alpha_{(m,k)} = \alpha_0/(mT + k)^\sigma$ in F-ML-IRL (Algorithm 1), where $\alpha_0 > 0$ and $\sigma \in (0, 1)$ are constants and $M$ is the total number of dual aggregations, the following convergence results hold for F-ML-IRL:*

$$\frac{1}{M} \sum_{m=0}^{M-1} \mathbb{E}\left[ \left\| \log(\overline{\pi}_{(m,T)}) - \log(\pi_{\overline{\theta}_{(m,T)}}) \right\|_\infty \right] = \mathcal{O}\left( M^{-1} \gamma^{T-1} \right) + \mathcal{O}\left( M^{-\sigma} T^{1-\sigma} \right), \tag{18}$$

$$\frac{1}{M} \sum_{m=0}^{M-1} \mathbb{E}\left[ \left\| \nabla L(\overline{\theta}_{(m,T)}) \right\|^2 \right] = \mathcal{O}(M^{-1}) + \mathcal{O}(M^{-\sigma} T^{-\sigma}) + \mathcal{O}(M^{-1-\sigma} T^{1-\sigma}). \tag{19}$$

**Remarks:** The time complexity of both policy estimate and reward parameter optimization depends on the number of global aggregation rounds $M$ and the number of local training steps $T$. The policy and reward function parameters in F-ML-IRL converge at the rate of $M^{-\sigma}T^{1-\sigma}$ and $M^{-\sigma}T^{-\sigma}$, respectively, since we have $\sigma \in (0,1)$ and $T$ is often fixed. We note that due to dual-aggregation and the variance caused by local training on distributed datasets across decentralized clients, F-ML-IRL exhibits a slightly slower convergence rate, compared with standard centralized ML-IRL with a single client (whose convergence rate is $M^{-\sigma}$). From Equations (18) and (19), there exists a sweet spot with respect to the number of local training steps $T$, since $\gamma^{T-1}$ and $T^{-\sigma}$ both descreases with $T$, while $T^{1-\sigma}$ increases. Exploring this tradeoff will be considered in future work. Compared with Fed-Avg (whose convergence rate is $M^{-1}T^{-1}$), F-ML-IRL also has a slower convergence rate due to the complexity of the bi-level optimization problem.

**Proof Sketch:** Due to space limitations, we outline the key steps of our convergence analysis and present the complete proof in the appendix. We first analyze the convergence of policy estimates and reduce it to the convergence of Q-values. We then analyze the distance between Q-values using the Lipschitz property, tracing back to the start of each dual aggregation around. In particular, we examine the extra distance between the estimated policy and the optimal policy caused by aggregation, seeking the contraction property of Q-value estimates between adjacent aggregation rounds. Next, for reward optimization, we leverage the Lipschitz smooth property of the likelihood and control the discrepancy between the stochastic gradient and the true gradient. This allows us to use the convergence of Q-values from the previous analysis to demonstrate the gradient convergence of the reward parameter. For simplicity of notations, we use $Q^{\text{soft}}_{i,(m,t)}$ to denote $Q^{\text{soft}}_{r_{\theta^i_{(m,t)}}, \pi^i_{(m,t)}}$, the action-value function at a given state for the local policy and reward parameter estimations at round $(m,t)$. Similarly, $Q^{\text{soft}}_{i,(m,t)}*$ denotes $Q^{\text{soft}}_{r_{\theta^i_{(m,t)}}, \pi_{\theta^i_{(m,t)}}}$, which is the Q-function for the optimal policy under the reward parameter at round $(m,t)$ and $Q^{\text{soft}}_{\overline{(m,t)}}$ denotes $Q^{\text{soft}}_{r_{\overline{\theta}^i_{(m,T)}}, \pi_{\overline{\theta}^i_{(m,T)}}}$, which represents the Q-function for the aggregated policy and reward parameter at the $m$'th aggregation.

**Convergence of Policy Estimate:**

**Step 1:** We show the distance between the aggregated policy and the optimal policy under the aggregated reward parameter could be controlled using the distance between soft Q-functions:

$$\| \log(\overline{\pi}_{(m,T)}) - \log(\pi_{\theta^i_{(m,T-1)}}) \|_\infty \leq 2\|\overline{Q}^{\text{soft}}_{(m,T-1)} - Q^{\text{soft}}_{i,(m,T-2)}*\|_\infty$$

This step relies on the policy update rule in equation 6.

**Step 2:** By introducing intermediary terms as bridges, specifically looking back to the time right after the last aggregation, where all local servers have the same reward parameter $\overline{\theta}_{(m-1,T)}$, we further bound the difference in **Step 1** by converting it to the difference of reward parameters using equation 9, 10, 15. Combining this with the $\gamma$-contraction property of the soft-Q update, we have:

$$\|\overline{Q}^{\text{soft}}_{(m,T-1)} - Q^{\text{soft}}_{i,(m,T-1)}*\|_\infty \leq \gamma^{T-2}\|Q^{\text{soft}}_{r_{\overline{\theta}_{(m-1,T)}}, \overline{\pi}_{(m-1,T)}} - Q^{\text{soft}}_{r_{\overline{\theta}_{(m-1,T)}}, \pi_{\overline{\theta}_{(m-1,T)}}}\|_\infty + E_1$$

where we use auxiliary variable $E_1 = 4\alpha\left(\frac{1-\gamma^{T-2}}{1-\gamma} + T - 2\right)L_q^2$.

**Step 3:** Using Lemma 2, we bound the difference in Q-values orresponding to different policies with the same reward during the aggregation step, and finally have:

$$\|\overline{Q}^{\text{soft}}_{(m,T-1)} - Q^{\text{soft}}_{i,(m,T-1)}*\|_\infty \leq (1-\gamma)\gamma^{T-1}\|\overline{Q}^{\text{soft}}_{(m-1,T-1)} - Q^{\text{soft}}_{i,(m-1,T-1)}*\|_\infty + E_2$$

where we use auxiliary variable $E_2 = 2\frac{1-\gamma^{T-2}}{1-\gamma} + (1-\gamma)^2\gamma^{T-2} + \frac{1-\gamma}{\gamma}(2T-3) + 2(T-2)L_q^2$. Finally, we obtain the convergence rate of the policy estimate by the contraction of Q-difference.

**Convergence of Reward Parameter Optimization:**

**Step 1:** We first leverage the Lipschitz smooth property of $l(\theta)$ equation 16:

$$L(\overline{\theta}_{(m,T)})) \geq L(\overline{\theta}_{(m-1,T)}) + \left\langle \nabla L(\overline{\theta}_{(m,T)}), \overline{\theta}_{(m,T)} - \overline{\theta}_{(m-1,T)} \right\rangle - \frac{L_c}{2}\|\overline{\theta}_{(m,T)} - \overline{\theta}_{(m-1,T)}\|^2$$

**Step 2:** We show the bias between the stochastic gradient estimate $g^i_{(m,k)}$ and the true gradient $\nabla L(\theta_{(m-1,T)})$ could be controlled. In this process, we also compare the increments of local clients

to control the extra error terms introduced by the federated scheme leveraging equation 9, 12. We show that the gradient of the global reward parameter could be bounded by the distance between Q-values:

$$\alpha(T-1)\mathbb{E}[\|\nabla L(\overline{\theta}_{(m-1,T)})\|^2] \le \alpha C_1 \mathbb{E}[\|\overline{Q}^{\text{soft}}_{(m-1,T-1)} - Q^{\text{soft}}_{i,(m-1,T-2)}\|_\infty] \\ + \mathbb{E}[L(\overline{\theta}_{(m,T)}) - L(\overline{\theta}_{(m-1,T)})] + E_3 \tag{20}$$

where $C_1 = \frac{4(1-\gamma^{T-1})}{\gamma} L_q^2 C_d \sqrt{|\mathcal{S}| \cdot |\mathcal{A}|}$ and $E_3 = 8\alpha L_q^3 C_d \sqrt{|\mathcal{S}| \cdot |\mathcal{A}|} \cdot \frac{T-1-\frac{1-\gamma^{T-1}}{1-\gamma}}{1-\gamma} + \frac{(T-1)(3T-1)\alpha^2 L_c L_q^2}{2} + \frac{4(1-\gamma^{T-1})}{1-\gamma}\alpha L_q^2 C_d \sqrt{|\mathcal{S}| \cdot |\mathcal{A}|} \cdot [2(2T-3)\alpha L_q^2 + \frac{1-\gamma}{\gamma} \cdot 4\alpha L_q^2]$ are two auxiliary variables. By combining this with the convergence of the Q-value difference that was established in **Step 3** of the Policy Estimation proof, we obtain the desired convergence of the reward parameter.

## 5 EVALUATIONS

We evaluated the proposed F-ML-IRL method on five high-dimensional robotic control tasks in MuJoCo Todorov et al. (2012). For comparison, we selected several state-of-the-art baselines, including imitation learning approaches that only learn the expert policy—specifically like BC Pomerleau (1988) and GAIL Ho & Ermon (2016), as well as IRL methods that simultaneously learn both a reward function and a policy, namely f-IRL Ni et al. (2021) and ML-IRL Zeng et al. (2022). To ensure fairness, we used Soft Actor-Critic (SAC) Haarnoja et al. (2018) as the base RL algorithm for all methods since it incorporates elements of Soft Q-Learning and achieves strong performance using the actor-critic scheme. The experiments are conducted on a server with AMD EPYC 7513 32-Core Processors and NVIDIA RTX A6000 GPUs. We choose $M = 200$ rounds and $T = 5$ local steps and average the results over multiple runs. Our evaluation code is available at https://anonymous.4open.science/r/F-ML-IRL/.

| Environment | Setting | F-ML-IRL | ML-IRL | | BC | | GAIL | | f-IRL | |
|---|---|---|---|---|---|---|---|---|---|---|
| | | | Mixed | Medium | Mixed | Medium | Mixed | Medium | Mixed | Medium |
| Ant | (3, 200) | **6425.91** | 6219.78 | 6161.65 | 983.99 | 984.04 | 989.30 | 988.73 | 5615.33 | 5930.89 |
| | (3, 1000) | **6398.98** | 5100.25 | 6402.87 | 5952.08 | 718.87 | 989.00 | 989.29 | 5370.28 | 5527.63 |
| | (5, 200) | **6254.32** | 5614.91 | 6161.65 | 983.51 | 984.04 | 988.67 | 988.73 | 5628.94 | 5930.89 |
| | (5, 1000) | **6528.04** | 6330.67 | 6402.87 | 411.83 | 718.87 | 989.77 | 989.29 | 5388.74 | 5527.63 |
| HalfCheetah | (3, 200) | **13007.75** | 8054.94 | 12581.28 | -0.63 | -0.73 | 7513.31 | 10288.42 | 10110.73 | 12962.52 |
| | (3, 1000) | 13228.98 | **13642.82** | 13124.24 | -0.66 | -11.74 | 12112.99 | 11506.59 | 13075.95 | 12871.64 |
| | (5, 200) | 11827.91 | 6406.45 | 12581.28 | -0.57 | -0.73 | 4910.45 | 10288.42 | 7132.01 | **12962.52** |
| | (5, 1000) | 12360.60 | 12750.04 | **13124.24** | 110.59 | -11.74 | 11364.40 | 11506.59 | 12659.30 | 12871.64 |
| Hopper | (3, 200) | 3576.10 | 1871.83 | **3623.07** | 18.11 | 18.13 | 1022.90 | 1023.65 | 1297.25 | 3456.47 |
| | (3, 1000) | **3674.64** | 3518.04 | 3479.33 | 18.17 | 2290.58 | 1025.93 | 1032.07 | 3403.36 | 3390.08 |
| | (5, 200) | 3419.95 | 1484.52 | **3623.07** | 18.12 | 18.13 | 1020.19 | 1023.65 | 1313.12 | 3456.47 |
| | (5, 1000) | **3618.44** | 3601.40 | 3479.33 | 1016.31 | 2290.58 | 1111.08 | 1032.07 | 3468.72 | 3390.08 |
| Humanoid | (3, 200) | 5656.06 | 5484.99 | 5861.01 | 243.21 | 242.50 | 4666.38 | 3035.34 | 5510.06 | **6004.58** |
| | (3, 1000) | 5694.79 | **5903.42** | 5813.57 | 241.46 | 532.41 | 4627.15 | 4688.41 | 5708.21 | 5726.86 |
| | (5, 200) | **6232.37** | 5462.64 | 5861.01 | 242.69 | 242.50 | 4692.86 | 3035.34 | 5523.40 | 6004.58 |
| | (5, 1000) | **6294.25** | 5713.37 | 5813.57 | 545.01 | 532.41 | 4577.12 | 4688.41 | 5608.83 | 5726.86 |
| Walker2d | (3, 200) | 4057.25 | 3317.61 | 4400.43 | 8.38 | 8.27 | 353.66 | 18.74 | 1050.78 | **5729.55** |
| | (3, 1000) | **5798.37** | 5061.23 | 5673.49 | 8.27 | 507.69 | 344.55 | 19.27 | 4805.53 | 5255.57 |
| | (5, 200) | 4540.90 | 3024.14 | 4400.43 | 8.40 | 8.27 | 13.03 | 18.74 | 1115.37 | **5729.55** |
| | (5, 1000) | **5853.42** | 4669.73 | 5673.49 | 711.06 | 507.69 | 360.10 | 19.27 | 4704.77 | 5255.57 |
| Average | – | **6712.60** | 5661.64 | 6712.09 | 575.97 | 529.00 | 3183.64 | 3359.05 | 5424.58 | 6685.58 |

Table 1: Compare F-ML-IRL and baselines on MuJoCo tasks, with different number of clients and demonstration trajectory length. F-ML-IRL achieves similar or higher recovered reward in almost all scenarios and outperforms the baselines in more than half, as well as in terms of the average.

We evaluate different algorithms using the rewards associated with the recovered expert policies evaluated in the original environment (same as the method adopted in previous work). We compare F-ML-IRL with the baselines on five MuJoCo tasks under non-iid data distributions, where each

client contains different demonstration data corresponding to varying levels of expertise. For the baselines that rely on centralized learning, we consider two setups: (i) a single client with medium-level demonstrations, denoted as *medium* and (ii) a single client with a mixture of demonstrations of different levels, denoted as *mixed*. In either case, the total amount of local data per client remains the same in the experiments. More details on experiment set up is provided in the appendix.

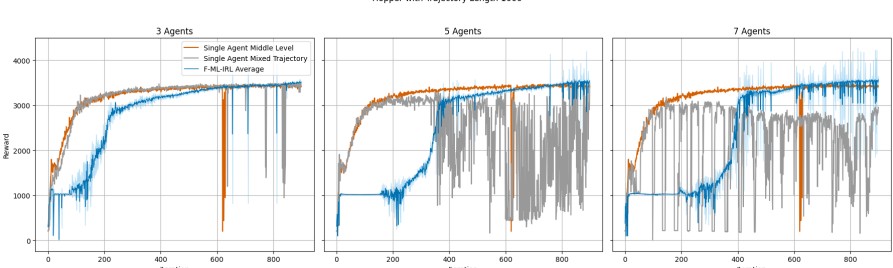

Figure 2: Convergence of F-ML-IRL in Hopper Environment compared with centralized ML-IRL with mixed and medium data. As the number of clients (and thus the non-iid datasets) increases from 3 (left) to 7 (right), F-ML-IRL takes longer to converge and nevertheless achieves more significant improvement by leveraging distributed demonstration data on the clients.

The evaluation results are summarized in Table 1. We have tested each algorithm and each MuJoCo task under 4 settings, i.e., with 3 or 5 clients and with demonstration trajectory length equals to 200 or 1000, respectively. As demonstrated in Zeng et al. (2022), even a single expert trajectory of length 1000 can lead to a well-recovered policy using ML-IRL. To investigate the performance of our model under conditions of scarce and distributed data, we utilize a single expert trajectory of length 1000 and further reduce its length to 200 in the experiments. In Table 1, we also compute the average reward for each algorithm across all settings and tasks in our experiments.

We note that F-ML-IRL ensures convergences of the recovered reward in decentralized learning and achieves similar or higher recovered reward than the baselines in almost all settings and tasks. It even outperforms centralized baselines in more than half of the settings and tasks, due to its ability to utilize distributed data. The performance of F-ML-IRL is pretty robust as the number of clients increases to 5 and the expert trajectory length reduces to 200. Imitation learning baselines like BC and GAIL generally have lower performance and even fail in some settings. While ML-IRL performs generally well, it fails to recover a satisfactory policy when data is limited or in tasks involving mixed trajectories of different expertise. On the other hand, f-IRL performs relatively well when provided with longer expert trajectories but struggles when demonstration data is limited. In contrast, our F-ML-IRL consistently achieves similar or higher recovered rewards compared to all baselines, particularly maintaining robust performance even when data is limited and involves demonstrations of mixed expertise.

We further illustrate the convergence of our F-ML-IRL algorithm compared with two different centralized learning baselines using ML-IRL (with medium and mixed-data, respectively) in the Hopper environment, as shown in Figure 2. As the number of clients (and thus the number of non-iid local datasets) increases (from 3 clients on the left to 7 clients on the right), it takes F-ML-IRL more rounds to converge, because of the increased variance introduced by local training on more participating clients and datasets. Nevertheless, F-ML-IRL is able to converge to higher recovered reward than both baselines. Centralized ML-IRL suffers with mixed demonstration data of varying expertise. In contrast, as the number of clients and demonstration dataset increases, F-ML-IRL shows more significant improvement by leverage distributed demonstration data on the clients.

## 6 CONCLUSIONS

This paper proposes F-ML-IRL for federated maximum-likelihood inverse reinforcement learning. It enables decentralized learning of a shared latent reward function from distributed datasets and using decentralized clients. F-ML-IRL algorithm leverages a dual-aggregation to update the shared global model and performs bi-level local updates for inverse learning. We analyze the convergence and time-complexity of F-ML-IRL. Evaluation results on MuJoCo tasks how that F-ML-IRL ensures convergences of the recovered reward and achieves similar or higher recovered reward, compared to state-of-the-art baselines using centralized inverse learning. For further work, we plan to investigate further communication reduction and the use of heterogeneous local models in F-ML-IRL.

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
