## A  PROOF OF LEMMA 2

Given the definition of soft-Q function following Bellman equation:

$$Q_{r,\pi}^{\text{soft}}(s,a) = \mathbb{E}_{s'\sim P(\cdot|s,a)}\left[r(s,a) + \gamma\mathbb{E}_{a'\sim\pi(\cdot|s')}\left(Q_{r,\pi}^{\text{soft}}(s',a') - \log\pi(a'|s')\right)\right] \tag{21}$$

The difference between soft-Q values under policies $\pi$ and $\pi'$ is:

$$|Q_{r,\pi}^{\text{soft}}(s,a) - Q_{r,\pi'}^{\text{soft}}(s,a)| = \gamma\left|\mathbb{E}_{s'\sim P(\cdot|s,a)}\left[\mathbb{E}_{a'\sim\pi(\cdot|s')}\left(Q_{r,\pi}^{\text{soft}}(s',a') - \log\pi(a'|s')\right)\right]\right.$$
$$\left. - \mathbb{E}_{s'\sim P(\cdot|s,a)}\left[\mathbb{E}_{a'\sim\pi'(\cdot|s')}\left(Q_{r,\pi'}^{\text{soft}}(s',a') - \log\pi'(a'|s')\right)\right]\right| \tag{22}$$

Using the triangle inequality, we separate the terms in equation 22:

$$|Q_{r,\pi}^{\text{soft}}(s,a) - Q_{r,\pi'}^{\text{soft}}(s,a)| \leq \gamma\Bigg(\left|\mathbb{E}_{s'\sim P(\cdot|s,a)}\mathbb{E}_{a'\sim\pi(\cdot|s')}\left[Q_{r,\pi}^{\text{soft}}(s',a') - Q_{r,\pi'}^{\text{soft}}(s',a')\right]\right|$$
$$+ \left|\mathbb{E}_{s'\sim P(\cdot|s,a)}\mathbb{E}_{a'\sim\pi(\cdot|s')}\left[\log\pi(a'|s') - \log\pi'(a'|s')\right]\right|\Bigg) \tag{23}$$

For the first term in equation 22, we apply Jensen's inequality to the absolute value function:

$$\left|\mathbb{E}_{s'\sim P(\cdot|s,a)}\mathbb{E}_{a'\sim\pi(\cdot|s')}\left[Q_{r,\pi}^{\text{soft}}(s',a') - Q_{r,\pi'}^{\text{soft}}(s',a')\right]\right| \leq \sup_{s',a'}|Q_{r,\pi}^{\text{soft}}(s',a') - Q_{r,\pi'}^{\text{soft}}(s',a')| \tag{24}$$

Similarly, the second term in equation 22 involving the log policies is bounded as:

$$\left|\mathbb{E}_{s'\sim P(\cdot|s,a)}\mathbb{E}_{a'\sim\pi(\cdot|s')}\left[\log\pi(a'|s') - \log\pi'(a'|s')\right]\right| \leq \sup_{s',a'}|\log\pi(a'|s') - \log\pi'(a'|s')| \tag{25}$$

Thus, we combine the two terms in equation 22:

$$|Q_{r,\pi}^{\text{soft}}(s,a) - Q_{r,\pi'}^{\text{soft}}(s,a)| \leq \gamma\sup_{s',a'}|Q_{r,\pi}^{\text{soft}}(s',a') - Q_{r,\pi'}^{\text{soft}}(s',a')| + \gamma\sup_{s',a'}|\log\pi(a'|s') - \log\pi'(a'|s')| \tag{26}$$

Since soft-Q values depend recursively on future rewards, we apply this bound recursively over time. At $t = 1$, the same bound holds:

$$|Q_{r,\pi}^{\text{soft}}(s_1,a_1) - Q_{r,\pi'}^{\text{soft}}(s_1,a_1)| \leq \gamma\sup_{s_2,a_2}|Q_{r,\pi}^{\text{soft}}(s_2,a_2) - Q_{r,\pi'}^{\text{soft}}(s_2,a_2)|$$
$$+ \gamma\sup_{s',a'}|\log\pi(a'|s') - \log\pi'(a'|s')| \tag{27}$$

Substituting this into the previous equation, we get:

$$|Q_{r,\pi}^{\text{soft}}(s_0 = s, a_0 = a) - Q_{r,\pi'}^{\text{soft}}(s_0 = s, a_0 = a)|$$
$$\leq \gamma^2\sup_{s_2,a_2}|Q_{r,\pi}^{\text{soft}}(s_2,a_2) - Q_{r,\pi'}^{\text{soft}}(s_2,a_2)| + \gamma\sup_{s',a'}|\log\pi(a'|s') - \log\pi'(a'|s')|(1+\gamma) \tag{28}$$

Applying this recursively over $n$ steps:

$$
|Q_{r,\pi}^{\text{soft}}(s_0 = s, a_0 = a) - Q_{r,\pi'}^{\text{soft}}(s_0 = s, a_0 = a)|
$$

$$
\leq \gamma^n \sup_{s_n, a_n} |Q_{r,\pi}^{\text{soft}}(s_n, a_n) - Q_{r,\pi'}^{\text{soft}}(s_n, a_n)| + \sup_{s', a'} |\log \pi(a'|s') - \log \pi'(a'|s')| \sum_{k=0}^{n-1} \gamma^k \tag{29}
$$

As $n \to \infty$, the term $\gamma^n \sup_{s_n, a_n} |Q_{r,\pi}^{\text{soft}}(s_n, a_n) - Q_{r,\pi'}^{\text{soft}}(s_n, a_n)|$ tends to zero. The geometric series sum is $\sum_{k=0}^{\infty} \gamma^k = \frac{1}{1-\gamma}$. Thus, taking the infinity norm with respect to all $s$ and $a$, the final bound is:

$$
\|Q_{r,\pi}^{\text{soft}} - Q_{r,\pi'}^{\text{soft}}\|_\infty \leq \frac{\gamma}{1-\gamma} \|\log(\pi) - \log(\pi')\|_\infty \tag{30}
$$

## B    PROOF OF THEOREM 1

In this section we show the complete proof for the convergence results for the recovered global policies $\overline{\pi}_{(m,T)}$ and the global reward parameter $\overline{\theta}_{(m,T)}$ after $m$ communications with communication period $T$.

### B.1    CONVERGENCE OF POLICY ESTIMATE $\overline{\pi}_{(m,T)}$

We first analyze the approximation error between the logarithm of the synchronized policy $\log(\overline{\pi}_{(m,T)}(a|s))$ and the logarithm of the optimal policy corresponding to the previous local reward parameter $\log(\pi_{\theta_{(m,T-1)}^i}(a|s))$ for all $i$. Specifically, we aim to bound the difference:

$$
\left| \log(\overline{\pi}_{(m,T)}(a|s)) - \log(\pi_{\theta_{(m,T-1)}^i}(a|s)) \right|
$$

This difference represents the discrepancy between the synchronized policy after the $m$-th global aggregation and the optimal policy corresponding to the previous local reward parameter $\theta_{(m,T-1)}^i$.

We aim to show that the distance between the logarithms of the synchronized policy and the optimal policy can be bounded by the difference between their corresponding soft-Q values. Specifically, we want to bound:

$$
\left| \log(\overline{\pi}_{(m,T)}(a|s)) - \log(\pi_{\theta_{(m,T-1)}^i}(a|s)) \right| \leq \Delta_Q,
$$

where $\Delta_Q$ involves the difference between the soft-Q values $\overline{Q}_{(m,T-1)}^{\text{soft}}(s,a)$ and $Q_{r_{\theta_{(m,T-2)}^i}, \pi_{\theta_{(m,T-2)}^i}}^{\text{soft}}(s,a)$.

Recall that the policy is proportional to the exponential of the soft-Q value in equation 6. Thus, we can write:

$$
\log(\overline{\pi}_{(m,T)}(a|s)) = \log\left( \frac{\exp\left(\overline{Q}_{m,T-1}^{\text{soft}}(s,a)\right)}{\sum_{\tilde{a}} \exp\left(\overline{Q}_{(m,T-1)}^{\text{soft}}(s,\tilde{a})\right)} \right)
$$

$$
= \overline{Q}_{(m,T-1)}^{\text{soft}}(s,a) - \log\left( \sum_{\tilde{a}} \exp\left(\overline{Q}_{(m,T-1)}^{\text{soft}}(s,\tilde{a})\right) \right) \tag{31}
$$

Since $\log(\pi_{\theta_{(m,T-1)}^i}(a|s))$ is the optimal policy under reward parameter $\theta_{(m,T-1)}^i$, according to (Haarnoja et al., 2017), it has the form

$$
\pi_{\theta_{(m,T-1)}^i}(a|s) = \frac{\exp(Q_{r_{\theta_{(m,T-2)}^i}, \pi_{\theta_{(m,T-2)}^i}(a|s)}^{\text{soft}}(s,a))}{\sum_{\tilde{a}} \exp(Q_{r_{\theta_{(m,T-2)}^i}, \pi_{\theta_{(m,T-2)}^i}(a|s)}^{\text{soft}}(s,\tilde{a}))} \tag{32}
$$

Similarly, we have:

$$\log(\pi_{\theta^i_{(m,T-1)}}(a|s)) = Q^{\text{soft}}_{r_{\theta^i_{(m,T-2)}},\pi_{\theta^i_{(m,T-2)}}}(s,a) - \log\left(\sum_{\tilde{a}}\exp\left(Q^{\text{soft}}_{r_{\theta^i_{(m,T-2)}},\pi_{\theta^i_{(m,T-2)}}}(s,\tilde{a})\right)\right) \tag{33}$$

Subtracting the two expressions and use the triangle inequality, we can bound the absolute value of the difference by the sum of the absolute values:

$$\left|\log(\overline{\pi}_{(m,T)}(a|s)) - \log(\pi_{\theta^i_{(m,T-1)}}(a|s))\right|$$

$$= \left|\left[\overline{Q}^{\text{soft}}_{(m,T-1)}(s,a) - \log\left(\sum_{\tilde{a}}\exp\left(\overline{Q}^{\text{soft}}_{(m,T-1)}(s,\tilde{a})\right)\right)\right]\right.$$

$$\left. - \left[Q^{\text{soft}}_{r_{\theta^i_{(m,T-2)}},\pi_{\theta^i_{(m,T-2)}}}(s,a) - \log\left(\sum_{\tilde{a}}\exp\left(Q^{\text{soft}}_{r_{\theta^i_{(m,T-2)}},\pi_{\theta^i_{(m,T-2)}}}(s,\tilde{a})\right)\right)\right]\right|$$

$$= \left|\overline{Q}^{\text{soft}}_{(m,T-1)}(s,a) - Q^{\text{soft}}_{r_{\theta^i_{(m,T-2)}},\pi_{\theta^i_{(m,T-2)}}}(s,a)\right.$$

$$\left. - \left[\log\left(\sum_{\tilde{a}}\exp\left(\overline{Q}^{\text{soft}}_{(m,T-1)}(s,\tilde{a})\right)\right) - \log\left(\sum_{\tilde{a}}\exp\left(Q^{\text{soft}}_{r_{\theta^i_{(m,T-2)}},\pi_{\theta^i_{(m,T-2)}}}(s,\tilde{a})\right)\right)\right]\right|$$

$$\leq \left|\overline{Q}^{\text{soft}}_{(m,T-1)}(s,a) - Q^{\text{soft}}_{r_{\theta^i_{(m,T-2)}},\pi_{\theta^i_{(m,T-2)}}}(s,a)\right|$$

$$+ \left|\log\left(\sum_{\tilde{a}}\exp\left(\overline{Q}^{\text{soft}}_{(m,T-1)}(s,\tilde{a})\right)\right) - \log\left(\sum_{\tilde{a}}\exp\left(Q^{\text{soft}}_{r_{\theta^i_{(m,T-2)}},\pi_{\theta^i_{(m,T-2)}}}(s,\tilde{a})\right)\right)\right| \tag{34}$$

The second term in equation 34 involves the difference of logarithms of sums. We can bound it using properties of logarithms and the maximum difference of the soft-Q values.

We utilize the following inequality (as referenced in Equation 47 of (Zeng et al., 2022)):

$$\left|\log\left(\sum_{\tilde{a}}\exp(Q_1(s,\tilde{a}))\right) - \log\left(\sum_{\tilde{a}}\exp(Q_2(s,\tilde{a}))\right)\right| \leq \max_{\tilde{a}}|Q_1(s,\tilde{a}) - Q_2(s,\tilde{a})| \tag{35}$$

Applying equation 35, we get:

$$\left|\log\left(\sum_{\tilde{a}}\exp\left(\overline{Q}^{\text{soft}}_{(m,T-1)}(s,\tilde{a})\right)\right) - \log\left(\sum_{\tilde{a}}\exp\left(Q^{\text{soft}}_{r_{\theta^i_{(m,T-2)}},\pi_{\theta^i_{(m,T-2)}}}(s,\tilde{a})\right)\right)\right|$$

$$\leq \max_{\tilde{a}}\left|\overline{Q}^{\text{soft}}_{(m,T-1)}(s,\tilde{a}) - Q^{\text{soft}}_{r_{\theta^i_{(m,T-2)}},\pi_{\theta^i_{(m,T-2)}}}(s,\tilde{a})\right| \tag{36}$$

Combining the results from equation 34 and equation 36:

$$\left|\log(\overline{\pi}_{(m,T)}(a|s)) - \log(\pi_{\theta^i_{(m,T-1)}}(a|s))\right|$$

$$\leq \left|\overline{Q}^{\text{soft}}_{(m,T-1)}(s,a) - Q^{\text{soft}}_{r_{\theta^i_{(m,T-2)}},\pi_{\theta^i_{(m,T-2)}}}(s,a)\right| \tag{37}$$

$$+ \max_{\tilde{a}}\left|\overline{Q}^{\text{soft}}_{(m,T-1)}(s,\tilde{a}) - Q^{\text{soft}}_{r_{\theta^i_{(m,T-2)}},\pi_{\theta^i_{(m,T-2)}}}(s,\tilde{a})\right|$$

Taking the infinity norm on equation 37 gives:

$$\|\log(\overline{\pi}_{(m,T)}) - \log(\pi_{\theta^i_{(m,T-1)}})\|_\infty \leq 2\|\overline{Q}^{\text{soft}}_{(m,T-1)} - Q^{\text{soft}}_{r_{\theta^i_{(m,T-2)}},\pi_{\theta^i_{(m,T-2)}}}\|_\infty \tag{38}$$

By the definition of aggregation in equation 10, we have

$$\overline{Q}^{\text{soft}}_{(m,T-1)} = \frac{1}{N}\sum_{j=1}^{N} Q^{\text{soft}}_{r_{\theta^j_{(m,T-2)}},\pi^j_{(m,T-2)}} \tag{39}$$

Plug above definition into equation 38 and by triangle inequality:

$$\|\overline{Q}^{\text{soft}}_{(m,T-1)} - Q^{\text{soft}}_{r_{\theta^i_{(m,T-2)}},\pi_{\theta^i_{(m,T-2)}}}\|_\infty = \|\sum_{j=1}^{N}\frac{1}{N}(Q^{\text{soft}}_{r_{\theta^j_{(m,T-2)}},\pi^j_{(m,T-2)}} - Q^{\text{soft}}_{r_{\theta^i_{(m,T-2)}},\pi_{\theta^i_{(m,T-2)}}})\|_\infty$$

$$\leq \frac{1}{N}\sum_{j=1}^{N}\|Q^{\text{soft}}_{r_{\theta^j_{(m,T-2)}},\pi^j_{(m,T-2)}} - Q^{\text{soft}}_{r_{\theta^i_{(m,T-2)}},\pi_{\theta^i_{(m,T-2)}}}\|_\infty \tag{40}$$

Therefore, we move to analyse $\|Q^{\text{soft}}_{r_{\theta^j_{(m,T-2)}},\pi^j_{(m,T-2)}} - Q^{\text{soft}}_{r_{\theta^i_{(m,T-2)}},\pi_{\theta^i_{(m,T-2)}}}\|_\infty$, which is the difference of soft-Q values between two different local nodes, one under policy estimation, and the other under optimal policy. Looking back to the time right after last aggregation, where all local servers have the same reward parameter $\overline{\theta}_{(m-1,T)}$, we could further bound this difference using the difference of reward parameters, since the difference of local reward parameters are introduced by the local increment at each internal iteration except for the aggregation round.

We start by decomposing this difference into three terms and use the triangle inequality, we bound the sum :

$$\left\|Q^{\text{soft}}_{r_{\theta^j_{(m,T-2)}},\pi^j_{(m,T-2)}} - Q^{\text{soft}}_{r_{\theta^i_{(m,T-2)}},\pi_{\theta^i_{(m,T-2)}}}\right\|_\infty$$

$$= \left\|\left(Q^{\text{soft}}_{r_{\theta^j_{(m,T-2)}},\pi^j_{(m,T-2)}} - Q^{\text{soft}}_{r_{\theta^j_{(m,T-2)}},\pi_{\theta^j_{(m,T-2)}}}\right)\right.$$
$$+ \left(Q^{\text{soft}}_{r_{\theta^j_{(m,T-2)}},\pi_{\theta^j_{(m,T-2)}}} - Q^{\text{soft}}_{r_{\theta^j_{(m,0)}},\pi_{\theta^j_{(m,0)}}}\right)$$
$$+ \left.\left(Q^{\text{soft}}_{r_{\theta^i_{(m,0)}},\pi_{\theta^i_{(m,0)}}} - Q^{\text{soft}}_{r_{\theta^i_{(m,T-2)}},\pi_{\theta^i_{(m,T-2)}}}\right)\right\|_\infty \tag{41}$$

$$\leq \left\|Q^{\text{soft}}_{r_{\theta^j_{(m,T-2)}},\pi^j_{(m,T-2)}} - Q^{\text{soft}}_{r_{\theta^j_{(m,T-2)}},\pi_{\theta^j_{(m,T-2)}}}\right\|_\infty$$
$$+ \left\|Q^{\text{soft}}_{r_{\theta^j_{(m,T-2)}},\pi_{\theta^j_{(m,T-2)}}} - Q^{\text{soft}}_{r_{\theta^j_{(m,0)}},\pi_{\theta^j_{(m,0)}}}\right\|_\infty$$
$$+ \left\|Q^{\text{soft}}_{r_{\theta^i_{(m,0)}},\pi_{\theta^i_{(m,0)}}} - Q^{\text{soft}}_{r_{\theta^i_{(m,T-2)}},\pi_{\theta^i_{(m,T-2)}}}\right\|_\infty$$

The first term is the difference between the soft-Q values under the same reward parameter $\theta^j_{(m,T-2)}$ but different policies $\pi^j_{(m,T-2)}$ and $\pi_{\theta^j_{(m,T-2)}}$. The second term is the difference due to the change in reward parameters from $\theta^j_{(m,T-2)}$ to $\theta^j_{(m,0)}$, with corresponding optimal policies, and the third term is similar to the second term but for node $i$, comparing $\theta^i_{(m,0)}$ and $\theta^i_{(m,T-2)}$. We are utilizing the fact that $\theta^j_{(m,0)} = \theta^i_{(m,0)}$ since they are initialized after previous aggregation.

Applying equation 15 to equation 41, we have:

$$\|Q^{\text{soft}}_{r_{\theta^j_{(m,T-2)}}, \pi^j_{(m,T-2)}} - Q^{\text{soft}}_{r_{\theta^i_{(m,T-2)}}, \pi_{\theta^i_{(m,T-2)}}}\|_\infty$$

$$\leq \left\|Q^{\text{soft}}_{r_{\theta^j_{(m,T-2)}}, \pi^j_{(m,T-2)}} - Q^{\text{soft}}_{r_{\theta^j_{(m,T-2)}}, \pi_{\theta^j_{(m,T-2)}}}\right\|_\infty \tag{42}$$

$$+ L_q \left\|\theta^j_{(m,T-2)} - \theta^j_{(m,0)}\right\| + L_q \left\|\theta^i_{(m,0)} - \theta^i_{(m,T-2)}\right\|$$

Next, we express the differences in reward parameters in terms of gradient updates. According to equation 9:

$$\theta^j_{(m,T-2)} = \theta^j_{(m,0)} + \alpha \sum_{k=0}^{T-3} g^j_{(m,k)} \tag{43}$$

$$\theta^i_{(m,T-2)} = \theta^i_{(m,0)} + \alpha \sum_{k=0}^{T-3} g^i_{(m,k)} \tag{44}$$

Substituting back into our equation 42:

$$\|Q^{\text{soft}}_{r_{\theta^j_{(m,T-2)}}, \pi^j_{(m,T-2)}} - Q^{\text{soft}}_{r_{\theta^i_{(m,T-2)}}, \pi_{\theta^i_{(m,T-2)}}}\|_\infty$$

$$\leq \left\|Q^{\text{soft}}_{r_{\theta^j_{(m,T-2)}}, \pi^j_{(m,T-2)}} - Q^{\text{soft}}_{r_{\theta^j_{(m,T-2)}}, \pi_{\theta^j_{(m,T-2)}}}\right\|_\infty + L_q\alpha \left\|\sum_{k=0}^{T-3} g^j_{(m,k)}\right\| + L_q\alpha \left\|\sum_{k=0}^{T-2} g^i_{(m,k)}\right\| \tag{45}$$

According to equation (56) in (Zeng et al., 2022)), the gradients are bounded as:

$$\|g^i_{(m,k)}\| \leq 2L_q \tag{46}$$

we can further bound the sums in equation 45:

$$\|Q^{\text{soft}}_{r_{\theta^j_{(m,T-2)}}, \pi^j_{(m,T-2)}} - Q^{\text{soft}}_{r_{\theta^i_{(m,T-2)}}, \pi_{\theta^i_{(m,T-2)}}}\|_\infty$$

$$\leq \left\|Q^{\text{soft}}_{r_{\theta^j_{(m,T-2)}}, \pi^j_{(m,T-2)}} - Q^{\text{soft}}_{r_{\theta^j_{(m,T-2)}}, \pi_{\theta^j_{(m,T-2)}}}\right\|_\infty + 4(T-2)\alpha L_q^2 \tag{47}$$

Now the difference is bounded by the difference between Q-values under reward parameter $\theta^j_{(m,T-2)}$ with respect to the previous approximated policy $\pi^j_{(m,T-2)}$ and the optimal policy $\pi_{\theta^j_{(m,T-2)}}$, plus some error terms. We come back from comparing Q-values across different nodes to evaluating the soft-Q value approximation within a single node.

By equation 57 in (Zeng et al., 2022):

$$\|Q^{\text{soft}}_{r_{\theta^i_{(m,k)}}, \pi^i_{(m,k)}} - Q^{\text{soft}}_{r_{\theta^i_{(m,k)}}, \pi_{\theta^i_{(m,k)}}}\|_\infty \leq \gamma \|Q^{\text{soft}}_{r_{\theta^i_{(m,k-1)}}, \pi^i_{(m,k-1)}} - Q^{\text{soft}}_{r_{\theta^i_{(m,k-1)}}, \pi_{\theta^i_{(m,k-1)}}}\|_\infty$$
$$+ 4\alpha L_q^2, \quad 1 \leq k \leq T-1, \quad m \in \mathbb{N}, \quad \forall i \tag{48}$$

Above inequality provides a bounds of the local Q-value using the previous Q-value times a contraction factor $\gamma$ plus some extra term. We could use it to compare the aggregated Q-value at m-th outer round with the aggregated Q-value at $m-1$ - th outer round:

$$\|Q^{\text{soft}}_{r_{\theta^i_{(m,k)}},\pi^i_{(m,k)}} - Q^{\text{soft}}_{r_{\theta^i_{(m,k)}},\pi_{\theta^i_{(m,k)}}}\|_\infty$$

$$\leq \gamma^k \|Q^{\text{soft}}_{r_{\theta^i_{(m,0)}},\pi^i_{(m,0)}} - Q^{\text{soft}}_{r_{\theta^i_{(m,0)}},\pi_{\theta^i_{(m,0)}}}\|_\infty + \sum_{i=0}^{k-1}\gamma^i \cdot 4\alpha L_q^2 \tag{49}$$

$$= \gamma^k \|Q^{\text{soft}}_{r_{\theta^i_{(m,0)}},\pi^i_{(m,0)}} - Q^{\text{soft}}_{r_{\theta^i_{(m,0)}},\pi_{\theta^i_{(m,0)}}}\|_\infty + \frac{1-\gamma^k}{1-\gamma}\cdot 4\alpha L_q^2, \quad m\in\mathbb{N}, \quad 1\leq k \leq T-1$$

Apply equation 49 to equation 47:

$$\|Q^{\text{soft}}_{r_{\theta^j_{(m,T-2)}},\pi^j_{(m,T-2)}} - Q^{\text{soft}}_{r_{\theta^i_{(m,T-2)}},\pi_{\theta^i_{(m,T-2)}}}\|_\infty$$

$$\leq \|Q^{\text{soft}}_{r_{\theta^j_{(m,T-2)}},\pi^j_{(m,T-2)}} - Q^{\text{soft}}_{r_{\theta^j_{(m,T-2)}},\pi_{\theta^j_{(m,T-2)}}}\|_\infty + 4(T-2))\alpha L_q^2$$

$$\leq \gamma^{T-2}\|Q^{\text{soft}}_{r_{\theta^j_{(m,0)}},\pi^j_{(m,0)}} - Q^{\text{soft}}_{r_{\theta^j_{(m,0)}},\pi_{\theta^j_{(m,0)}}}\|_\infty + \frac{1-\gamma^{T-2}}{1-\gamma}\cdot 4\alpha L_q^2 + 4(T-2)\alpha L_q^2$$

$$= \gamma^{T-2}\|Q^{\text{soft}}_{r_{\overline{\theta}_{(m-1,T)}},\overline{\pi}_{(m-1,T)}} - Q^{\text{soft}}_{r_{\overline{\theta}_{(m-1,T)}},\pi_{\overline{\theta}_{(m-1,T)}}}\|_\infty + 4\alpha\left(\frac{1-\gamma^{T-2}}{1-\gamma}+T-2\right)L_q^2 \tag{50}$$

Plug equation 50 into equation 40:

$$\|\overline{Q}^{\text{soft}}_{(m,T-1)} - Q^{\text{soft}}_{r_{\theta^i_{(m,T-2)}},\pi_{\theta^i_{(m,T-2)}}}\|_\infty$$

$$\leq \frac{1}{N}\sum_{j=1}^N \|Q^{\text{soft}}_{r_{\theta^j_{(m,T-2)}},\pi^j_{(m,T-2)}} - Q^{\text{soft}}_{r_{\theta^i_{(m,T-1)}},\pi_{\theta^i_{(m,T-1)}}}\|_\infty$$

$$\leq \frac{1}{N}\sum_{j=1}^N \left[\gamma^{T-2}\|Q^{\text{soft}}_{r_{\overline{\theta}_{(m-1,T)}},\overline{\pi}_{(m-1,T)}} - Q^{\text{soft}}_{r_{\overline{\theta}_{(m-1,T)}},\pi_{\overline{\theta}_{(m-1,T)}}}\|_\infty + 4\alpha\left(2\cdot\frac{1-\gamma^{T-2}}{1-\gamma}+T-2\right)L_q^2\right]$$

$$= \gamma^{T-2}\|Q^{\text{soft}}_{r_{\overline{\theta}_{(m-1,T)}},\overline{\pi}_{(m-1,T)}} - Q^{\text{soft}}_{r_{\overline{\theta}_{(m-1,T)}},\pi_{\overline{\theta}_{(m-1,T)}}}\|_\infty + 4\alpha\left(\frac{1-\gamma^{T-2}}{1-\gamma}+T-2\right)L_q^2 \tag{51}$$

We further analyze $\|Q^{\text{soft}}_{r_{\overline{\theta}_{(m-1,T)}},\overline{\pi}_{(m-1,T)}} - Q^{\text{soft}}_{r_{\overline{\theta}_{(m-1,T)}},\pi_{\overline{\theta}_{(m-1,T)}}}\|_\infty$ and use triangle inequality to decompose it into three parts to acquire the same form of Q-value difference in the previous outer round:

$$\|Q^{\text{soft}}_{r_{\overline{\theta}_{(m-1,T)}},\overline{\pi}_{(m-1,T)}} - Q^{\text{soft}}_{r_{\overline{\theta}_{(m-1,T)}},\pi_{\overline{\theta}_{(m-1,T)}}}\|_\infty$$

$$= \|(Q^{\text{soft}}_{r_{\overline{\theta}_{(m-1,T)}},\overline{\pi}_{(m-1,T)}} - \overline{Q}^{\text{soft}}_{(m-1,T-1)}) + (\overline{Q}^{\text{soft}}_{(m-1,T-1)} - Q^{\text{soft}}_{r_{\theta^i_{(m-1,T-1)}},\pi_{\theta^i_{(m-1,T-1)}}})$$

$$+ (Q^{\text{soft}}_{r_{\theta^i_{(m-1,T-1)}},\pi_{\theta^i_{(m-1,T-1)}}} - Q^{\text{soft}}_{r_{\overline{\theta}_{(m-1,T)}},\pi_{\overline{\theta}_{(m-1,T)}}})\|_\infty$$

$$\leq \|(Q^{\text{soft}}_{r_{\overline{\theta}_{(m-1,T)}},\overline{\pi}_{(m-1,T)}} - \overline{Q}^{\text{soft}}_{(m-1,T-1)}\|_\infty + \|\overline{Q}^{\text{soft}}_{(m-1,T-1)} - Q^{\text{soft}}_{r_{\theta^i_{(m-1,T-1)}},\pi_{\theta^i_{(m-1,T-1)}}}\|_\infty$$

$$+ \|Q^{\text{soft}}_{r_{\theta^i_{(m-1,T-1)}},\pi_{\theta^i_{(m-1,T-1)}}} - Q^{\text{soft}}_{r_{\overline{\theta}_{(m-1,T)}},\pi_{\overline{\theta}_{(m-1,T)}}}\|_\infty \tag{52}$$

We first bound the first term in equation 52 by introducing an middle term and triangle inequality:

$$\|Q^{\text{soft}}_{r_{\overline{\theta}_{(m-1,T)}},\overline{\pi}_{(m-1,T)}} - \overline{Q}^{\text{soft}}_{(m-1,T-1)}\|_\infty$$

$$= \|\overline{Q}^{\text{soft}}_{(m-1,T-1)} - \frac{1}{N}\sum_{j=1}^N Q^{\text{soft}}_{r_{\theta^j_{(m-1,T-2)}},\pi^j_{(m-1,T-2)}}\|_\infty$$

$$= \|(Q^{\text{soft}}_{r_{\overline{\theta}_{(m-1,T)}},\overline{\pi}_{(m-1,T)}} - Q^{\text{soft}}_{r_{\theta^j_{(m-1,T-2)}},\overline{\pi}_{(m-1,T)}})$$

$$+ (Q^{\text{soft}}_{r_{\theta^j_{(m-1,T-2)}},\overline{\pi}_{(m-1,T)}} - \frac{1}{N}\sum_{j=1}^N Q^{\text{soft}}_{r_{\theta^j_{(m-1,T-2)}},\pi^j_{(m-1,T-2)}})\| \tag{53}$$

$$\leq \|Q^{\text{soft}}_{r_{\overline{\theta}_{(m-1,T)}},\overline{\pi}_{(m-1,T)}} - Q^{\text{soft}}_{r_{\theta^j_{(m-1,T-2)}},\overline{\pi}_{(m-1,T)}}\|_\infty$$

$$+ \frac{1}{N}\sum_{j=1}^N \|Q^{\text{soft}}_{r_{\theta^j_{(m-1,T-2)}},\overline{\pi}_{(m-1,T)}} - Q^{\text{soft}}_{r_{\theta^j_{(m-1,T-2)}},\pi^j_{(m-1,T-2)}}\|_\infty$$

For the first term in equation 53, we leverage Lemma 7 in (Zeng et al., 2022), which states:

$$|Q^{\text{soft}}_{r_{\theta_1},\pi} - Q^{\text{soft}}_{r_{\theta_2},\pi}| \leq L_q \|\theta_1 - \theta_2\|, \forall \pi, \forall \theta_1, \theta_2, \forall s \in \mathcal{S}, \forall a \in \mathcal{A} \tag{54}$$

Then we could further bound the difference between $\theta$'s using equation 9 and equation 46:

$$\|Q^{\text{soft}}_{r_{\overline{\theta}_{(m-1,T)}},\overline{\pi}_{(m-1,T)}} - Q^{\text{soft}}_{r_{\theta^j_{(m-1,T-2)}},\overline{\pi}_{(m-1,T)}}\|_\infty$$

$$\leq L_q \|\overline{\theta}_{(m-1,T)} - \theta^j_{(m-1,T-2)}\|_\infty$$

$$= L_q \|\frac{1}{N}\sum_{j=1}^N \theta^j_{(m-1,T-1)} - \theta^j_{(m-1,T-2)}\|$$

$$\leq \frac{L_q}{N}\sum_{j=1}^N \|\theta^j_{(m-1,T-1)} - \theta^i_{(m-1,T-2)}\|$$

$$= \frac{L_q}{N}\sum_{j=1}^N \|(\theta^j_{(m-1,T-1)} - \theta^j_{(m-1,0)}) + (\theta^i_{(m-1,0)} - \theta^i_{(m-1,T-2)})\|$$

$$\leq \frac{L_q}{N}\sum_{j=1}^N (\|\theta^j_{(m-1,T-1)} - \theta^j_{(m-1,0)}\| + \|\theta^i_{(m-1,0)} - \theta^i_{(m-1,T-2)}\|)$$

$$= \frac{L_q}{N}\sum_{j=1}^N \left(\alpha \left\|\sum_{k=0}^{T-2} g^j_{(m,k)}\right\| + \alpha \left\|\sum_{k=0}^{T-3} g^i_{(m,k)}\right\|\right)$$

$$\leq 2(2T-3)\alpha L_q^2$$

For the second term in equation 53, by Lemma 2:

$$\|Q^{\text{soft}}_{r_{\theta^j_{(m-1,T-2)}},\overline{\pi}_{(m-1,T)}} - Q^{\text{soft}}_{r_{\theta^j_{(m-1,T-2)}},\pi^j_{(m-1,T-2)}}\|_\infty$$

$$\leq \frac{1-\gamma}{\gamma}\|\log(\overline{\pi}_{(m-1,T)}) - \log(\pi^j_{(m-1,T-2)})\|_\infty \tag{55}$$

We get result similar to equation 38 using similar techniques and decompose equation 55 using triangle inequality:

$$\| \log(\overline{\pi}_{(m-1,T)}) - \log(\pi^j_{(m-1,T-2)}) \|_\infty$$

$$\leq 2 \| \overline{Q}^{\text{soft}}_{(m-1,T-1)} - Q^{\text{soft}}_{r_{\theta^i_{(m-1,T-3)}}, \pi_{\theta^i_{(m-1,T-3)}}} \|_\infty$$

$$= 2 \| (\overline{Q}^{\text{soft}}_{(m-1,T-1)} - Q^{\text{soft}}_{r_{\theta^i_{(m-1,T-2)}}, \pi_{\theta^i_{(m-1,T-2)}}}) + (Q^{\text{soft}}_{r_{\theta^i_{(m-1,T-2)}}, \pi_{\theta^i_{(m-1,T-2)}}} - Q^{\text{soft}}_{r_{\theta^i_{(m-1,T-3)}}, \pi_{\theta^i_{(m-1,T-3)}}}) \|_\infty$$

$$\leq 2 (\| \overline{Q}^{\text{soft}}_{(m-1,T-1)} - Q^{\text{soft}}_{r_{\theta^i_{(m-1,T-2)}}, \pi_{\theta^i_{(m-1,T-2)}}} \|_\infty + \| Q^{\text{soft}}_{r_{\theta^i_{(m-1,T-2)}}, \pi_{\theta^i_{(m-1,T-2)}}} - Q^{\text{soft}}_{r_{\theta^i_{(m-1,T-3)}}, \pi_{\theta^i_{(m-1,T-3)}}} \|_\infty)$$

$$(56)$$

The last term could be controlled according to what we did for the last two terms in equation 41:

$$\| Q^{\text{soft}}_{r_{\theta^i_{(m-1,T-2)}}, \pi_{\theta^i_{(m-1,T-2)}}} - Q^{\text{soft}}_{r_{\theta^i_{(m-1,T-3)}}, \pi_{\theta^i_{(m-1,T-3)}}} \|_\infty \leq 2\alpha L_q^2 \tag{57}$$

Plugging above results into equation 52, we have:

$$\| Q^{\text{soft}}_{r_{\overline{\theta}_{(m-1,T)}}, \overline{\pi}_{(m-1,T)}} - Q^{\text{soft}}_{r_{\overline{\theta}_{(m-1,T)}}, \pi_{\overline{\theta}_{(m-1,T)}}} \|_\infty$$

$$\leq \frac{1-\gamma}{\gamma} \| \overline{Q}^{\text{soft}}_{(m-1,T-1)} - Q^{\text{soft}}_{r_{\theta^i_{(m-1,T-2)}}, \pi_{\theta^i_{(m-1,T-2)}}} \|_\infty + 2(2T-3)\alpha L_q^2 + \frac{1-\gamma}{\gamma} \cdot 4\alpha L_q^2 \tag{58}$$

We further plug equation 58 to equation 51:

$$\| \overline{Q}^{\text{soft}}_{(m,T-1)} - Q^{\text{soft}}_{r_{\theta^i_{(m,T-2)}}, \pi_{\theta^i_{(m,T-2)}}} \|_\infty$$

$$\leq \gamma^{T-2} \| Q^{\text{soft}}_{r_{\overline{\theta}_{(m-1,T)}}, \overline{\pi}_{(m-1,T)}} - Q^{\text{soft}}_{r_{\overline{\theta}_{(m-1,T)}}, \pi_{\overline{\theta}_{(m-1,T)}}} \|_\infty + 4\alpha \left( \frac{1-\gamma^{T-2}}{1-\gamma} + T - 2 \right) L_q^2$$

$$\leq (1-\gamma)\gamma^{T-1} \| \overline{Q}^{\text{soft}}_{(m-1,T-1)} - Q^{\text{soft}}_{r_{\theta^i_{(m-1,T-2)}}, \pi_{\theta^i_{(m-1,T-2)}}} \|_\infty$$

$$+ \left( 2\frac{1-\gamma^{T-2}}{1-\gamma} + (1-\gamma)^2\gamma^{T-2} + \frac{1-\gamma}{\gamma}(2T-3) + 2(T-2) \right) L_q^2 \tag{59}$$

Summing the inequality from $m = 1$ to $m = M$ gives:

$$\sum_{m=1}^{M} \| \overline{Q}^{\text{soft}}_{(m,T-1)} - Q^{\text{soft}}_{r_{\theta^i_{(m,T-2)}}, \pi_{\theta^i_{(m,T-2)}}} \|_\infty$$

$$\leq (1-\gamma)\gamma^{T-1} \sum_{m=1}^{M} \| \overline{Q}^{\text{soft}}_{(m-1,T-1)} - Q^{\text{soft}}_{r_{\theta^i_{(m-1,T-2)}}, \pi_{\theta^i_{(m-1,T-2)}}} \|_\infty$$

$$+ M \cdot [2\frac{1-\gamma^{T-2}}{1-\gamma} + (1-\gamma)^2\gamma^{T-2} + \frac{1-\gamma}{\gamma}(2T-3) + 2(T-2)]L_q^2 \tag{60}$$

Rearranging the inequality, it holds that:

$$\left(1 - (1-\gamma)\gamma^{T-1}\right) \sum_{m=1}^{M} \|\overline{Q}^{\text{soft}}_{(m,T-1)} - Q^{\text{soft}}_{r_{\theta^i_{(m,T-2)}}, \pi_{\theta^i_{(m,T-2)}}}\|_\infty$$

$$\leq (1-\gamma)\gamma^{T-1} \left( \|\overline{Q}^{\text{soft}}_{(0,T-1)} - Q^{\text{soft}}_{r_{\theta^i_{(0,T-2)}}, \pi_{\theta^i_{(0,T-2)}}}\|_\infty \right.$$

$$\left. - \|\overline{Q}^{\text{soft}}_{(M,T-1)} - Q^{\text{soft}}_{r_{\theta^i_{(M,T-2)}}, \pi_{\theta^i_{(M,T-2)}}}\|_\infty \right)$$

$$+ M \cdot [2\frac{1-\gamma^{T-2}}{1-\gamma} + (1-\gamma)^2\gamma^{T-2} + \frac{1-\gamma}{\gamma}(2T-3) + 2(T-2)]L_q^2$$

$$\leq (1-\gamma)\gamma^{T-1} \left( \|\overline{Q}^{\text{soft}}_{(0,T-1)} - Q^{\text{soft}}_{r_{\theta^i_{(0,T-2)}}, \pi_{\theta^i_{(0,T-2)}}}\|_\infty \right.$$

$$+ M \cdot [2\frac{1-\gamma^{T-2}}{1-\gamma} + (1-\gamma)^2\gamma^{T-2} + \frac{1-\gamma}{\gamma}(2T-3) + 2(T-2)]L_q^2$$

$$\tag{61}$$

Dividing by $1 - (1-\gamma)\gamma^{T-1}$ on both sides, we get

$$\sum_{m=1}^{M} \|\overline{Q}^{\text{soft}}_{(m,T-1)} - Q^{\text{soft}}_{r_{\theta^i_{(m,T-2)}}, \pi_{\theta^i_{(m,T-2)}}}\|_\infty$$

$$\leq \frac{(1-\gamma)\gamma^{T-1}}{1 - (1-\gamma)\gamma^{T-1}} \|\overline{Q}^{\text{soft}}_{(0,T-1)} - Q^{\text{soft}}_{r_{\theta^i_{(0,T-2)}}, \pi_{\theta^i_{(0,T-2)}}}\|_\infty$$

$$+ M \cdot \frac{2\frac{1-\gamma^{T-2}}{1-\gamma} + (1-\gamma)^2\gamma^{T-2} + \frac{1-\gamma}{\gamma}(2T-3) + 2(T-2)]L_q^2}{1 - (1-\gamma)\gamma^{T-1}}$$

Denote $C_0 = \|\overline{Q}^{\text{soft}}_{(0,T-1)} - Q^{\text{soft}}_{r_{\theta^i_{(0,T-2)}}, \pi_{\theta^i_{(0,T-2)}}}\|_\infty$.

Dividing by $M$ on both sides, we get

$$\frac{1}{M} \sum_{m=1}^{M} \|\overline{Q}^{\text{soft}}_{(m,T-1)} - Q^{\text{soft}}_{r_{\theta^i_{(m,T-2)}}, \pi_{\theta^i_{(m,T-2)}}}\|_\infty$$

$$\leq \frac{(1-\gamma)\gamma^{T-1}}{(1 - (1-\gamma)\gamma^{T-1})M} C_0 + \frac{2\frac{1-\gamma^{T-2}}{1-\gamma} + (1-\gamma)^2\gamma^{T-2} + \frac{1-\gamma}{\gamma}(2T-3) + 2(T-2)]L_q^2}{1 - (1-\gamma)\gamma^{T-1}} \tag{62}$$

Recall the step size is defined as $\alpha_{(m,t)} = \frac{\alpha_0}{(mT+t)^\sigma}$ where $\sigma > 0$. Then we have the following result:

$$\frac{1}{M} \sum_{m=1}^{M} \|\overline{Q}^{\text{soft}}_{(m,T-1)} - Q^{\text{soft}}_{r_{\theta^i_{(m,T-2)}}, \pi_{\theta^i_{(m,T-2)}}}\|_\infty$$

$$= \mathcal{O}(M^{-1}\gamma^T) + \mathcal{O}(M^{-\sigma}T^{1-\sigma}) \tag{63}$$

Going back to the convergence of policy approximation:

$$\frac{1}{M} \sum_{m=1}^{M} |\log(\overline{\pi}_{(m,T)}(a|s)) - \log(\pi_{\theta^i_{(m,T-1)}}(a|s))|_\infty$$

$$= \mathcal{O}(M^{-1}\gamma^T) + \mathcal{O}(M^{-\sigma}T^{1-\sigma}) \tag{64}$$

## B.2 CONVERGENCE OF THE GLOBAL REWARD PARAMETER $\overline{\theta}_{(m,T)}$

By the Lipschitz smooth property of the likelihood target equation 16, the definition of reward aggregation equation 12, and the reward parameter update rule equation 9:

$$
\begin{aligned}
&L(\overline{\theta}_{(m,T)})) \\
&\geq L(\overline{\theta}_{(m-1,T)}) + \left\langle \nabla L(\overline{\theta}_{(m,T)}), \overline{\theta}_{(m,T)} - \overline{\theta}_{(m-1,T)} \right\rangle - \frac{L_c}{2}\|\overline{\theta}_{(m,T)} - \overline{\theta}_{(m-1,T)}\|^2 \\
&= L(\overline{\theta}_{(m-1,T)}) + \left\langle \nabla L(\overline{\theta}_{(m,T)}), \frac{1}{N}\sum_{j=1}^{N}(\theta_{(m,T-1)}^j - \theta_{(m,0)}^j) \right\rangle - \frac{L_c}{2}\|\frac{1}{N}\sum_{j=1}^{N}(\theta_{(m,T-1)}^j - \theta_{(m,0)}^j)\|^2 \\
&= L(\overline{\theta}_{(m-1,T)}) + \alpha\left\langle \nabla L(\overline{\theta}_{(m,T)}), \frac{1}{N}\sum_{j=1}^{N}\sum_{k=0}^{T-2} g_{(m,k)}^j \right\rangle - \frac{L_c\alpha^2}{2}\|\frac{1}{N}\sum_{j=1}^{N}\sum_{k=0}^{T-2} g_{(m,k)}^j\|^2
\end{aligned}
\tag{65}
$$

We compare $g_{(m,k)}^i$ with the true gradient of $L(\theta_{(m,k)}^i)$ and leverage the fact that $\|\nabla L(\theta)\|_\infty \leq 2L_q$:

$$
\begin{aligned}
&L(\overline{\theta}_{(m,T)})) \\
&\geq L(\overline{\theta}_{(m-1,T)}) + \alpha\left\langle \nabla L(\overline{\theta}_{(m,T)}), \frac{1}{N}\sum_{j=1}^{N}\sum_{k=0}^{T-2} g_{(m,k)}^j \right\rangle - \frac{L_c\alpha^2}{2}\|\frac{1}{N}\sum_{j=1}^{N}\sum_{k=0}^{T-2} g_{(m,k)}^j\|^2 \\
&\geq L(\overline{\theta}_{(m-1,T)}) + \alpha\left\langle \nabla L(\overline{\theta}_{(m,T)}), \frac{1}{N}\sum_{j=1}^{N}\sum_{k=0}^{T-2} g_{(m,k)}^j - (T-1)\nabla L(\overline{\theta}_{(m-1,T)}) \right\rangle \\
&\quad + \alpha(T-1)\|\nabla L(\overline{\theta}_{(m-1,T)})\|^2 - \frac{L_c\alpha^2}{2}\|\frac{1}{N}\sum_{j=1}^{N}\sum_{k=0}^{T-2} g_{(m,k)}^j\|^2 \\
&\geq L(\overline{\theta}_{(m-1,T)}) - 2\alpha L_q \cdot \|\frac{1}{N}\sum_{j=1}^{N}\sum_{k=0}^{T-2}[g_{(m,k)}^j - \nabla L(\overline{\theta}_{(m-1,T)})]\| \\
&\quad + \alpha(T-1)\|\nabla L(\overline{\theta}_{(m-1,T)})\|^2 - (T-1)^2 \cdot \frac{L_c L_q^2\alpha^2}{2}
\end{aligned}
\tag{66}
$$

For $g^j(m,k)$ in equation 66, we evaluate its distance to $\nabla L(\theta_{(m,k)}^j)$ and also consider the distance between $\nabla L(\theta_{(m,k)}^j)$ and $\nabla L(\overline{\theta}_{(m,T)})$ with the help of triangle inequality:

$$
\begin{aligned}
&L(\overline{\theta}_{(m,T)})) \\
&\geq L(\overline{\theta}_{(m-1,T)}) - 2\alpha Lq \cdot \|\frac{1}{N}\sum_{j=1}^{N}\sum_{k=0}^{T-2}[g_{(m,k)}^j - \nabla L(\theta_{(m,k)}^j) + \nabla L(\theta_{(m,k)}^j) - \nabla L(\theta_{(m,0)}^j)]\| \\
&\quad + \alpha(T-1)\|\nabla L(\overline{\theta}_{(m-1,T)})\|^2 - \frac{(T-1)^2 L_c L_q^2\alpha^2}{2} \\
&\geq L(\overline{\theta}_{(m-1,T)}) - 2\alpha L_q \cdot [\frac{1}{N}\sum_{j=1}^{N}\sum_{k=0}^{T-2}(\|g_{(m,k)}^j - \nabla L(\theta_{(m,k)}^j)\| + \|\nabla L(\theta_{(m,k)}^j) - \nabla L(\theta_{(m,0)}^j)\|)] \\
&\quad + \alpha(T-1)\|\nabla L(\overline{\theta}_{(m-1,T)})\|^2 - \frac{(T-1)^2 L_c L_q^2\alpha^2}{2}
\end{aligned}
\tag{67}
$$

Taking expectation over both sides:

$$\mathbb{E}[L(\overline{\theta}_{(m,T)})]$$

$$\geq \mathbb{E}[L(\overline{\theta}_{(m-1,T)})] - 2\alpha L_q \cdot [\frac{1}{N}\sum_{j=1}^{N}\sum_{k=0}^{T-2}(\mathbb{E}\|g_{(m,k)}^{j} - \nabla L(\theta_{(m,k)}^{j})\| + \|\nabla L(\theta_{(m,k)}^{j}) - \nabla L(\theta_{(m,0)})\|)]$$

$$+ \alpha(T-1)\|\nabla L(\overline{\theta}_{(m-1,T)})\|^2 - \frac{(T-1)^2 L_c L_q^2 \alpha^2}{2}$$

$$(68)$$

According to equation (62) and (63) in (Zeng et al., 2022), we have:

$$\mathbb{E}\|g_{(m,k)}^{j} - \nabla L(\theta_{(m,k)}^{j})\| \leq 2L_q C_d \sqrt{|\mathcal{S}| \cdot |\mathcal{A}|} \mathbb{E}[\|Q_{r_{\theta_{(m,k)}^{j}},\pi_{(m,k)}^{j}}^{\text{soft}} - Q_{r_{\theta_{(m,k)}^{j}},\pi_{\theta_{(m,k)}^{j}}}^{\text{soft}}\|_\infty \quad (69)$$

Then, using the Lipschitz property of $L$ in equation 16:

$$\mathbb{E}[L(\overline{\theta}_{(m,T)})]$$

$$\geq \mathbb{E}[L(\overline{\theta}_{(m-1,T)})] - 2\alpha L_q \cdot \frac{1}{N}\sum_{j=1}^{N}\sum_{k=0}^{T-2}(2L_q C_d \sqrt{|\mathcal{S}| \cdot |\mathcal{A}|} \mathbb{E}[\|Q_{r_{\theta_{(m,k)}^{j}},\pi_{(m,k)}^{j}}^{\text{soft}} - Q_{r_{\theta_{(m,k)}^{j}},\pi_{\theta_{(m,k)}^{j}}}^{\text{soft}}\|_\infty$$

$$+ \mathbb{E}[\|\nabla L(\theta_{(m,k)}^{j}) - \nabla L(\theta_{(m,0)}^{j})\|]) + \alpha(T-1)\mathbb{E}[\|\nabla L(\overline{\theta}_{(m-1,T)})\|^2] - \frac{(T-1)^2 L_c L_q^2 \alpha^2}{2}$$

$$\geq \mathbb{E}[L(\theta_{(m-1,T)}^{i})] - 2\alpha L_q \cdot [\frac{1}{N}\sum_{j=1}^{N}\sum_{k=0}^{T-2}(2L_q C_d \sqrt{|\mathcal{S}| \cdot |\mathcal{A}|} \mathbb{E}[\|Q_{r_{\theta_{(m,k)}^{j}},\pi_{(m,k)}^{j}}^{\text{soft}} - Q_{r_{\theta_{(m,k)}^{j}},\pi_{\theta_{(m,k)}^{j}}}^{\text{soft}}\|_\infty]$$

$$+ \mathbb{E}[L_c\|\theta_{(m,k)}^{j} - \theta_{(m,0)}^{j}\|]) + \alpha(T-1)\mathbb{E}[\|\nabla L(\overline{\theta}_{(m-1,T)})\|^2] - \frac{(T-1)^2 L_c L_q^2 \alpha^2}{2}$$

$$(70)$$

Similar to equation 47, we have $\|\theta_{(m,k)}^{j} - \theta_{(m,0)}^{j}\| \leq 2k\alpha L_q$, applying equation 49 to equation 70, we have:

$$\mathbb{E}[L(\overline{\theta}_{(m,T)})]$$

$$\geq \mathbb{E}[L(\overline{\theta}_{(m-1,T)})] - 2\alpha L_q \cdot [\frac{1}{N}\sum_{j=1}^{N}\sum_{k=0}^{T-2}(2L_q C_d \sqrt{|\mathcal{S}| \cdot |\mathcal{A}|} \gamma^k \mathbb{E}[\|Q_{r_{\theta_{(m,0)}^{j}},\pi_{(m,0)}^{j}}^{\text{soft}} - Q_{r_{\theta_{(m,0)}^{j}},\pi_{\theta_{(m,0)}^{j}}}^{\text{soft}}\|_\infty]]$$

$$+ 2L_q C_d \sqrt{|\mathcal{S}| \cdot |\mathcal{A}|}\frac{1-\gamma^k}{1-\gamma} \cdot 4\alpha L_q^2 + 2k\alpha L_c L_q) + \alpha(T-1)\mathbb{E}[\|\nabla L(\overline{\theta}_{(m-1,T)})\|^2] - \frac{(T-1)^2 L_c L_q^2 \alpha^2}{2}$$

$$= \mathbb{E}[L(\overline{\theta}_{(m-1,T)})] - 2\alpha L_q \cdot \sum_{k=0}^{T-2}\gamma^k(2L_q C_d \sqrt{|\mathcal{S}| \cdot |\mathcal{A}|} \mathbb{E}[\|Q_{r_{\overline{\theta}_{(m-1,T)}},\overline{\pi}_{(m-1,T)}}^{\text{soft}} - Q_{r_{\overline{\theta}_{(m-1,T)}},\pi_{\overline{\theta}_{(m-1,T)}}}^{\text{soft}}\|_\infty]$$

$$+ 2L_q C_d \sqrt{|\mathcal{S}| \cdot |\mathcal{A}|}\frac{1-\gamma^k}{1-\gamma} \cdot 4\alpha L_q^2 + k\alpha L_c L_q) + \alpha(T-1)\mathbb{E}[\|\nabla L(\overline{\theta}_{(m-1,T)})\|^2] - \frac{(T-1)^2 L_c L_q^2 \alpha^2}{2}$$

$$= \mathbb{E}[L(\overline{\theta}_{(m-1,T)})] - \frac{4(1-\gamma^{T-1})}{1-\gamma}\alpha L_q^2 C_d \sqrt{|\mathcal{S}| \cdot |\mathcal{A}|} \mathbb{E}[\|Q_{r_{\overline{\theta}_{(m-1,T)}},\overline{\pi}_{(m-1,T)}}^{\text{soft}} - Q_{r_{\overline{\theta}_{(m-1,T)}},\pi_{\overline{\theta}_{(m-1,T)}}}^{\text{soft}}\|_\infty]$$

$$- 8\alpha L_q^3 C_d \sqrt{|\mathcal{S}| \cdot |\mathcal{A}|} \cdot \frac{T-1-\frac{1-\gamma^{T-1}}{1-\gamma}}{1-\gamma} - T(T-1)\alpha^2 L_c L_q^2 + \alpha(T-1)\mathbb{E}[\|\nabla L(\overline{\theta}_{(m-1,T)})\|^2]$$

$$- \frac{(T-1)^2 L_c L_q^2 \alpha^2}{2}$$

$$(71)$$

Plugging the result in equation 58, we have:

$$\mathbb{E}[L(\overline{\theta}_{(m,T)})]$$

$$\geq \mathbb{E}[L(\overline{\theta}_{(m-1,T)})] - \frac{4(1-\gamma^{T-1})}{1-\gamma}\alpha L_q^2 C_d\sqrt{|\mathcal{S}|\cdot|\mathcal{A}|}\mathbb{E}[\|Q^{\text{soft}}_{r_{\overline{\theta}_{(m-1,T)}},\overline{\pi}_{(m-1,T)}} - Q^{\text{soft}}_{r_{\overline{\theta}_{(m-1,T)}},\pi_{\overline{\theta}_{(m-1,T)}}}\|_\infty]$$

$$- 8\alpha L_q^3 C_d\sqrt{|\mathcal{S}|\cdot|\mathcal{A}|}\cdot\frac{T-1-\frac{1-\gamma^{T-1}}{1-\gamma}}{1-\gamma} - T(T-1)\alpha^2 L_c L_q^2 + \alpha(T-1)\mathbb{E}[\|\nabla L(\overline{\theta}_{(m-1,T)})\|^2]$$

$$- \frac{(T-1)^2 L_c L_q^2 \alpha^2}{2}$$

$$\geq \mathbb{E}[L(\overline{\theta}_{(m-1,T)})] - \frac{4(1-\gamma^{T-1})}{\gamma}\alpha L_q^2 C_d\sqrt{|\mathcal{S}|\cdot|\mathcal{A}|}\mathbb{E}[\|\overline{Q}^{\text{soft}}_{(m-1,T-1)} - Q^{\text{soft}}_{r_{\theta^i_{(m-1,T-2)}},\pi_{\theta^i_{(m-1,T-2)}}}\|_\infty]$$

$$+ \alpha(T-1)\mathbb{E}[\|\nabla L(\overline{\theta}_{(m-1,T)})\|^2] - 8\alpha L_q^3 C_d\sqrt{|\mathcal{S}|\cdot|\mathcal{A}|}\cdot\frac{T-1-\frac{1-\gamma^{T-1}}{1-\gamma}}{1-\gamma} - T(T-1)\alpha^2 L_c L_q^2$$

$$- \frac{(T-1)^2 L_c L_q^2 \alpha^2}{2} - \frac{4(1-\gamma^{T-1})}{1-\gamma}\alpha L_q^2 C_d\sqrt{|\mathcal{S}|\cdot|\mathcal{A}|}\cdot[2(2T-3)\alpha L_q^2 + \frac{1-\gamma}{\gamma}\cdot 4\alpha L_q^2] \tag{72}$$

Rearranging the inequality above and denote $C_1 = \frac{4(1-\gamma^{T-1})}{\gamma}L_q^2 C_d\sqrt{|\mathcal{S}|\cdot|\mathcal{A}|}$, we obtain:

$$\alpha(T-1)\mathbb{E}[\|\nabla L(\overline{\theta}_{(m-1,T)})\|^2]$$

$$\leq \alpha C_1\mathbb{E}[\|\overline{Q}^{\text{soft}}_{(m-1,T-1)} - Q^{\text{soft}}_{r_{\theta^i_{(m-1,T-2)}},\pi_{\theta^i_{(m-1,T-2)}}}\|_\infty] + \mathbb{E}[L(\overline{\theta}_{(m,T)}) - L(\overline{\theta}_{(m-1,T)})]$$

$$+ 8\alpha L_q^3 C_d\sqrt{|\mathcal{S}|\cdot|\mathcal{A}|}\cdot\frac{T-1-\frac{1-\gamma^{T-1}}{1-\gamma}}{1-\gamma} + \frac{(T-1)(3T-1)\alpha^2 L_c L_q^2}{2}$$

$$+ \frac{4(1-\gamma^{T-1})}{1-\gamma}\alpha L_q^2 C_d\sqrt{|\mathcal{S}|\cdot|\mathcal{A}|}\cdot[2(2T-3)\alpha L_q^2 + \frac{1-\gamma}{\gamma}\cdot 4\alpha L_q^2] \tag{73}$$

$$\leq \alpha C_1\mathbb{E}[\|\overline{Q}^{\text{soft}}_{(m-1,T-1)} - Q^{\text{soft}}_{r_{\theta^i_{(m-1,T-2)}},\pi_{\theta^i_{(m-1,T-2)}}}\|_\infty] + \mathbb{E}[L(\overline{\theta}_{(m,T)}) - L(\overline{\theta}_{(m-1,T)})]$$

$$+ 8\alpha L_q^3 C_d\sqrt{|\mathcal{S}|\cdot|\mathcal{A}|}\cdot\frac{T-1}{1-\gamma} + \frac{(T-1)(3T-1)\alpha^2 L_c L_q^2}{2}$$

$$+ \frac{16(T-1+\frac{1-\gamma}{\gamma})}{1-\gamma}\alpha^2 L_q^4 C_d\sqrt{|\mathcal{S}|\cdot|\mathcal{A}|}$$

Summing the inequality above from $m=1$ to $M$ and dividing both sides by $\alpha(T-1)M$, it holds that

$$\frac{1}{M}\sum_{m=1}^{M}\mathbb{E}[\|\nabla L(\overline{\theta}_{(m-1,T)})\|^2]$$

$$\leq \frac{1}{M(T-1)}\sum_{m=1}^{M}\mathbb{E}[\|\overline{Q}^{\text{soft}}_{(m-1,T-1)} - Q^{\text{soft}}_{r_{\theta^i_{(m-1,T-2)}},\pi_{\theta^i_{(m-1,T-2)}}}\|_\infty] + \mathbb{E}[\frac{L(\overline{\theta}_{(m,T)}) - L(\overline{\theta}_{(0,T)})}{\alpha(T-1)M}]$$

$$+ \frac{1}{M}\frac{8}{1-\gamma}L_q^3 C_d\sqrt{|\mathcal{S}|\cdot|\mathcal{A}|} + \frac{(3T-1)\alpha}{M}\frac{L_c L_q^2}{2} + \frac{16\alpha(1+\frac{1}{T-1}\frac{1-\gamma}{\gamma})}{M}\frac{L_q^4 C_d\sqrt{|\mathcal{S}|\cdot|\mathcal{A}|}}{1-\gamma} \tag{74}$$

Since $L(\overline{\theta}_{(m,T)})$ is negative and $L(\overline{\theta}_{(0,T)})$ is bounded constant, we plug equation 63 into equation 74 and get:

$$\frac{1}{M} \sum_{m=1}^{M} \mathbb{E}[\|\nabla L(\overline{\theta}_{(m-1,T)})\|^2]] = \mathcal{O}(M^{-1}) + \mathcal{O}(M^{-\sigma} T^{-\sigma}) + \mathcal{O}(M^{-1-\sigma} T^{1-\sigma}) \qquad (75)$$

## C  EVALUATION

We present the details of the experiment setup and show more convergence plots in MuJoCo tasks.

### C.1  EXPERIMENT SETUP

For f-IRL, we utilize the official implementation available at `https://github.com/twni2016/f-IRL`, which also includes implementations for BC and GAIL. The official implementation of ML-IRL can be found at `https://github.com/Cloud0723/ML-IRL`.

To ensure a fair comparison, we use SAC as the base RL algorithm for our F-ML-IRL approach as well as for all baselines, and Adam as the optimizer. Both the Q-network and policy network are configured as $64 \times 64$ MLPs with ReLU activation functions, and the learning rate is set to $1 \times 10^{-3}$.

For the Ant and Humanoid environments, the reward function is parameterized by a $128 \times 128$ MLP with ReLU activation, while for HalfCheetah, Hopper, and Walker2d, a $64 \times 64$ MLP with ReLU activation is used. The learning rate for the reward parameter is $1 \times 10^{-4}$ for Hopper and $1 \times 10^{-3}$ for the other environments.

At each iteration, we sample 10 trajectories from the current local policy estimate and compare them with the expert demonstration to update the reward parameter.

The reward levels of the expert demonstrations are shown in Table C.1. For 3 agents, we use Data 3, 4, and 5; for 5 agents, Data 2, 3, 4, 5, and 6 are used. For 7 agents, all 7 data sets are distributed across different local clients.

| Environment | Data 1 | Data 2 | Data 3 | Data 4 | Data 5 | Data 6 | Data 7 |
|---|---|---|---|---|---|---|---|
| Ant | 5465.10 | 5544.83 | 5699.96 | 5758.39 | 5820.41 | 5927.86 | 6035.14 |
| HalfCheetah | 12831.84 | 12973.62 | 13045.36 | 13187.47 | 13236.31 | 13328.38 | 13434.40 |
| Hopper | 3122.05 | 3217.36 | 3305.78 | 3424.81 | 3553.03 | 3603.60 | 3709.89 |
| Humanoid | 4934.42 | 5074.53 | 5134.65 | 5297.35 | 5345.38 | 5420.32 | 5501.73 |
| Walker2d | 4801.82 | 4976.41 | 5081.29 | 5193.50 | 5220.25 | 5379.48 | 5440.2 |

Table 2: The reward levels of 7 expert demonstration datasets that are used in our experiments across 5 MuJoCo tasks. We distribute these non-iid datasets to the clients in our experiments.

### C.2  CONVERGENCE PLOTS

We provide supplementary plots in other settings (different environment and trajectory length) here to show the convergence of F-ML-IRL compared with ML-IRL in two centralized learning data cases:

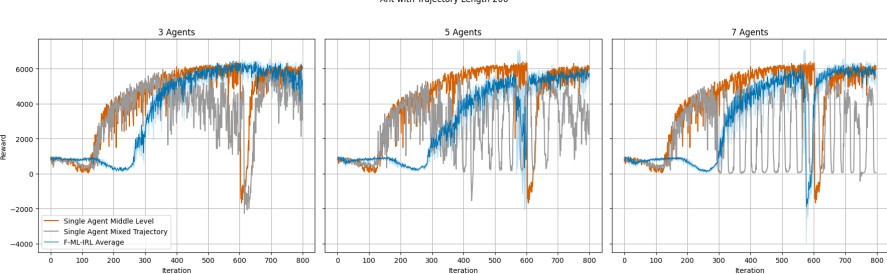

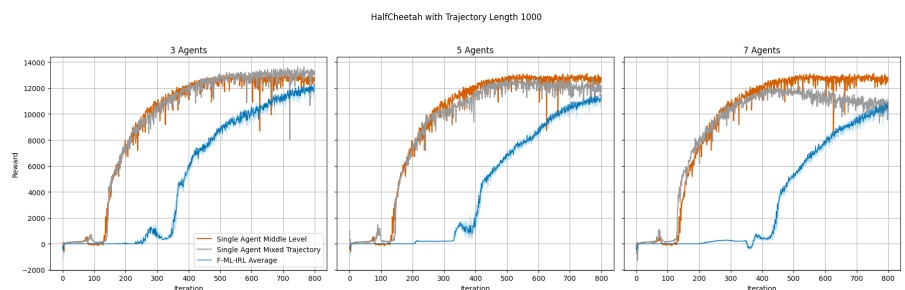

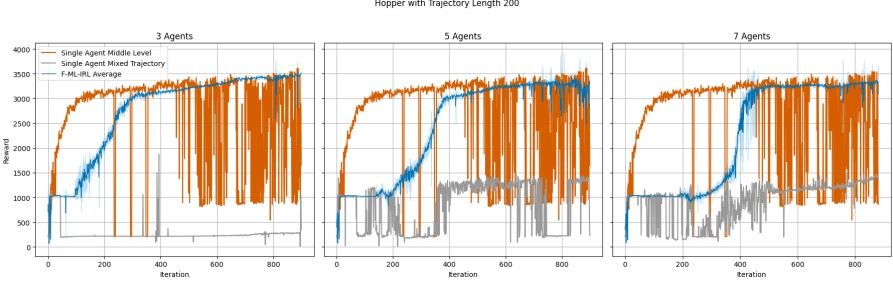

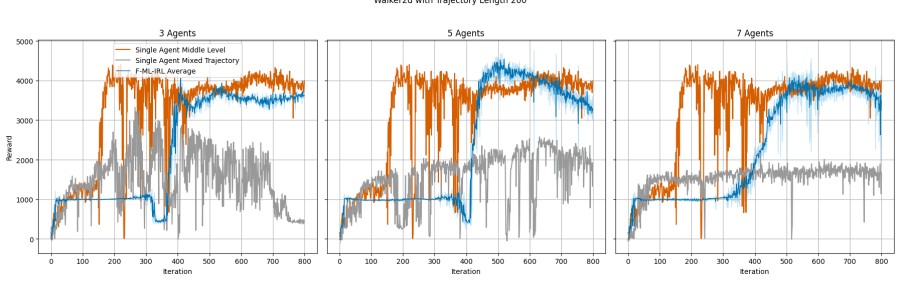

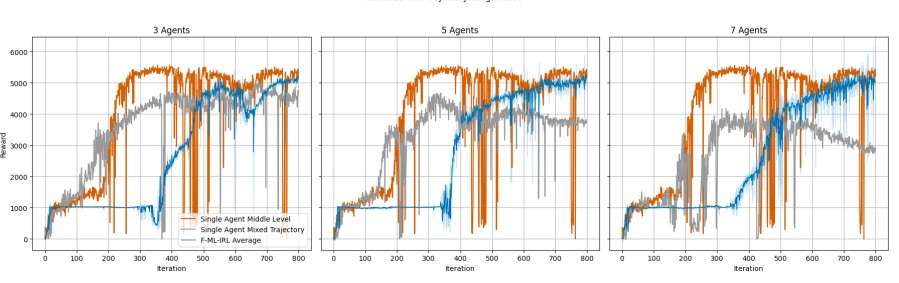