# OpenReview forum: "Federated Maximum Likelihood Inverse Reinforcement Learning with Convergence Guarantee"
_ICLR.cc/2025/Conference — Submitted to ICLR 2025_

### Official Review · Reviewer_QuNC · 2024-10-16

**Soundness:** 2
**Presentation:** 1
**Contribution:** 2
**Rating:** 3
**Confidence:** 4

**Summary:**

The authors extend the maximum-likelihood formulation for IRL to the federated setting, i.e., when data is shared across multiple devices and subject to privacy constraints. Then, the authors provide both a theoretical and empirical evaluation of the proposed algorithm.

**Strengths:**

- The authors provide both theoretical and empirical validation of the proposed algorithm.
- The analysed problem is novel and of practical importance.

**Weaknesses:**

PRESENTATION:

The paper is written in a very imprecise and gross manner. Specifically:
- Many symbols are introduced along the way since the background section in confusing and lacks some definitions.
- the related works are written very bad. It is not clear whether the authors have understood the content of some papers mentioned in the related works. For instance, in line 108 they say that "Apprenticeship learning via inverse reinforcement learning" use the recovered reward to derive an effective policy, while the authors do not recover any reward in that paper.
- No citation for lines 144-145. Moreover, what is $f$ in Eq. (3)?
- The preliminaries are also written very badly.

There are many typos. Some of them are:
- 045: FL never defined
- 107,109, ... : must use \citep instead of \citet
- 117: "an MDP"
- 285: "inequation"
- ... many others

CONTRIBUTION:
The contribution of the paper is fairly poor. While the idea is interesting, the problem is addressed in a non-adequate manner. Specifically:
- The paper is incremental: They simply take the algorithm developed by Zeng et al. (ML-IRL) and extend it in a very simple manner with some averages. While simplicity is in general an amazing tool, I believe that here it reflects a poor analysis of the problem.
- There is no technical novelty.

**Questions:**

- In Eq. (4), why not weighting on the number of trajectories owned by each client? Why the average?

---

> ### Author Response · Authors · 2024-11-30
>
> Thank you very much for your valuable comments and feedback. Below, we address the identified weaknesses and questions.
>
> **Weakness**
>
> - Presentation: Thank you for highlighting the presentation issues. We provide the following clarifications to address your concerns:
> In “Apprenticeship Learning via Inverse Reinforcement Learning,” the authors assume that the reward function is a linear combination of features and recover the weights of these features to define the reward function. This reward function aids in the apprenticeship learning process of finding an optimal policy.
> In our work,  f  in Eq. (3) refers to the objective function in the optimization problem of federated learning.
> Thank you very much for pointing out the typos, we have corrected them and updated the paper.
>
> **Questions**
>
> In our algorithm, the expert trajectories are sampled from the local trajectory databases. As a result, the number of trajectories contributing to model training is the same for each local client in every iteration. For simplicity, we chose to average across clients rather than apply weights based on the number of trajectories owned by each client. However, we acknowledge that this may not fully account for variance differences between clients. Moving forward, we plan to extend our approach by incorporating weighted averaging to address these differences and improve robustness.
>
> We sincerely appreciate your comments and hope these clarifications address your concerns.

---

### Official Review · Reviewer_AXzk · 2024-10-16

**Soundness:** 2
**Presentation:** 2
**Contribution:** 1
**Rating:** 3
**Confidence:** 4

**Summary:**

This paper proposes federated maximum likelihood IRL, an integration of federated learning and ML-IRL to handle the case where the demonstration data is distributed to a group of clients. The paper proposes an algorithm to solve this problem and provide finite-time convergence guarantee. MuJoCo simulations are used to validate the effectiveness of the proposed algorithm.

**Strengths:**

The topic on the integration of federated learning and IRL has some practical applications.

**Weaknesses:**

1. The main weakness is that the contribution is limited or is not clearly demonstrated. The paper seems to be a simple combination of federated learning and ML-IRL, and the current presentation largely follows ML-IRL. It is possible that there are some unique challenges caused by combining federated learning and ML-IRL, and these challenges do not appear in ML-IRL nor federated learning. However, the current presentation does not clearly demonstrate what the unique challenges are. I highly suggest the authors to outline the unique tenchincal challenges of this paper. From the current presentation, the problem formulation and the algorithm are simple combinations of federated learning and ML-IRL. For example, the problem formulation (4) is almost the same as the problem formulation in ML-IRL. The only difference is that the upper-level problem is now distributed. Also for the algorithm, lines 7-12 in Algorithm 1 are just the policy update and reward update of ML-IRL. Lines 17-19 are also standard practice in federated learning to aggregate parameters. Can the authors specify the unique challenges solved by this paper or some unique mechanisms proposed by this paper?

2. The discussion with the most relevant literature is lacking. In introduction (line 49), it is said that "We note that IRL using decentralized clients and distributed data is an open problem." However, there are already some distributed IRL works that can handle distributed data and distributed clients. The authors do not discuss the distinctions from these works, and even do not mention these works. By a simple search in Google with the keyword "distributed inverse reinforcement learning", I can find the following papers that solve for distributed demonstration data:

[1] Distributed inverse constrained reinforcement learning for multi-agent systems
[2] Distributed inverse optimal control
[3] Distributed estimation and inverse reinforcement learning for multi-agent systems

[1] and [2] both provide theoretical guarantees. I highly suggest the authors to compare with the existing distributed IRL methods given that they are the most relevant works to this paper.

3. The motivation of solving for distributed data needs to be better justified. The introduction (line 41) says that "demonstration data in practice are often distributed across decentralized clients". Why? Can the authors provide some concrete applications or scnarios to justify this?

4. It is said in related works (lines 159-160) that "However, existing FL methods could not be directly applied to the ML-IRL problem with decentralized clients, since ML-IRL requires a bi-level optimization involving both policy improvement and reward estimate using maximum likelihood." This sentence is also questionable. Basically, this sentence means that FL methods cannot be directly used because this problem is bi-level. However, there are already some theoretical papers on federated bi-level optimization (see below):

[4] Communication-Efficient Federated Bilevel Optimization with Local and Global Lower Level Problems

[5] Achieving Linear Speedup in Non-IID Federated Bilevel Learning

[6] FedNest: Federated Bilevel, Minimax, and Compositional Optimization

Can the authors discuss why these works cannot solve the problem? I highly suggest the authors to discuss the distinctions from the most relevant literature in detail, otherwise, it is hard to see the contribution of this paper.

**Questions:**

What is the unique challenge solved by this paper other than simply combining federated learning and ML-IRL? By unique challenges, I mean the challenges that do not appear in ML-IRL, federated learning, and distributed bi-level optimization.

---

> ### Author Response · Authors · 2024-11-30
>
> Thank you for your valuable feedback. Below, we address the identified weaknesses and questions.
>
> **Weakness 1**: The unique challenges addressed in this paper include the design of a federated learning algorithm tailored for maximum likelihood inverse reinforcement learning and the theoretical proof of its convergence guarantee. These contributions are central to our work and distinguish it from prior approaches. In our setup, the inner RL problem is formulated as soft Q-learning for Maximum Likelihood Inverse Reinforcement Learning. To ensure compatibility with this specific RL form, we decided to perform aggregation only on the Q-function rather than both the Q-function and the policy. This design choice simplifies the aggregation process and facilitates convergence.The most challenging part of the convergence proof lies in addressing the influence of aggregation. Our proof focuses on the tabular case, where the state and action spaces are finite. Prior ML-IRL work only considered a single agent problem and didn’t analyze it in a federated setting as how global aggregation impacts local models. On the other hand, traditional federated learning convergence analyses cannot be directly applied here because the lower-level problem is a soft Q-learning RL problem and we do not have a gradient-based update rule for the tabular case. Therefore, our construction of convergence is based on soft Q-functions and the entropy-regularized Boltzmann policy updates, which prevents us from utilizing existing proof techniques in the federated learning area. Facing above challenges, we carefully conduct analysis to bound the errors caused by the aggregation of Q-values and reward parameters to evaluate the optimality of the inner-loop RL problem with respect to the Q-values and the outer-loop maximum likelihood estimation.
>
> **Weakness 2**: Thank you for highlighting concerns regarding the literature review. While there are related topics discussed in [1], [2], and [3], their focuses are notably different:
> [1] addresses inverse constrained reinforcement learning,
> [2] explores inverse optimal control, and
> [3] investigates two-player zero-sum games.
> Each of these works employs fundamentally different methodologies and solutions compared to our approach. We believe that specific problems, such as federated inverse reinforcement learning, require targeted analysis that directly addresses the unique challenges of the domain.
>
> **Weakness 3**: In fields like cybersecurity and healthcare, demonstration data are often confidential and distributed across different locations. The data cannot be transmitted and collected for centralized IRL methods. This creates a significant challenge in data sharing due to privacy concerns and data scarcity. Our federated learning approach is designed to address these issues by enabling collaboration across distributed datasets without violating privacy constraints.
>
> **Weakness 4**: Thank you for raising this important question. While [4], [5], and [6] present general federated bilevel learning algorithms, the specific challenges of inverse reinforcement learning are distinct. Unlike general optimization problems, the low-level RL problem involves interaction with an environment and is characterized by sequential decision-making processes. These unique features necessitate targeted analysis, as different RL models require tailored solutions that consider the dynamics of their specific settings.
>
> **Questions**: For your question, please refer to our clarification under Weakness 1, as it directly addresses the uniqueness of our contributions in algorithm design and theoretical guarantees.
>
> We sincerely appreciate your comments and hope these clarifications address your concerns.

---

> > ### Comment · Reviewer_AXzk · 2024-11-30
> >
> > Thanks for the response. I really appreciate the authors' effort in addressing my comments.
> >
> >  The major contribution is the proof associated with the aggregation of Q-function. Is my understanding correct? If this is the case, can the authors elaborate the unique difference from standard federated learning literature since this aggregation mechanism is quite standard and the associated proof is widely studied in literature. What is the difference in proof for the aggreagtion part?
> >
> > For the reference [1,2], I disagree with the authors. In specific, [1] studies inverse constrained reinforcement learning that learns both reward and constraints. It can reduce to IRL (that only learns a reward) as a special case. [2] studies inverse optimal control, which is exactly IRL. In fact, IRL is originated from inverse optimal control.

---

> > > ### Author Response · Authors · 2024-12-03
> > >
> > > Dear Reviewer AXzk,
> > >
> > > Thank you for your feedback.
> > >
> > > For the proof, we leverage the Q-values and policy update rule to seek detailed analysis. With the RL problem involved, convergence analysis is different from traditional federated learning literature since we could not have a gradient-based update rule for both the Q-function and the policy. Therefore, tailored analysis has to be addressed for the specific form of the inner RL problem we choose for IRL and the outer loop associated with the reward function parameters. Our main contribution to the proof is that we follow the structure of this maximum likelihood IRL problem and provide targeted convergence analysis to this specific problem solution. See our clarification for weakness 1 for more details on the difficulties in the proof.
> > >
> > > Thank you again for your comments.
> > >
> > > Best,
> > > Authors of paper 10244

---

### Official Review · Reviewer_usqH · 2024-10-30

**Soundness:** 3
**Presentation:** 3
**Contribution:** 2
**Rating:** 5
**Confidence:** 4

**Summary:**

This paper develops an algorithm for federated inverse reinforcement learning (IRL). The algorithm combines maximum likelihood IRL in the reference [Zeng 2022] and model averaging in federated learning. Convergence analysis of the algorithm is provided. Experiments are conducted in MuJoCo where four baselines are included.

**Strengths:**

1. Federated IRL is an interesting addition to the literature of IRL.
2. Convergence of the algorithm is mathematically guaranteed.
3. The paper is clearly written and easy to follow.

**Weaknesses:**

1. The novelty of the algorithm is quite limited. As shown in Figure 1, the algorithm consists of local training and global update. The algorithm of the local training is identical to the maximum likelihood IRL algorithm in the reference [Zeng 2022]. The global update adopts averaging, a classic approach in federated learning.
2. The introduction claims that IRL with distributed data is an open problem. The statement is incorrect. Below are some recent papers on the topic.
R1: Federated Inverse Reinforcement Learning for Smart ICUs with Differential Privacy
R2: Federated Imitation Learning: A Privacy Considered Imitation Learning Framework for Cloud Robotic Systems with Heterogeneous Sensor Data
R3: Federated Imitation Learning: A Cross-Domain Knowledge Sharing Framework for Traffic Scheduling in 6G Ubiquitous IoT
R4: Federated Imitation Learning for UAV Swarm Coordination in Urban Traffic Monitoring
R5: Distributed Inverse Constrained Reinforcement Learning for Multi-agent Systems
3. The local Q functions and the global policy are communicated. Thus, the algorithm cannot deal with continuous state and action spaces. For large-scale discrete state and action spaces, the communication overhead is high.
4. The experiments should compare the developed algorithm with some algorithms in the above references. In addition, the centralized algorithms should have the same set of trajectories as the federated IRL algorithm. Consider the case that the federated IRL algorithm has 3 clients and each client has 100 trajectories. ML-IRL should be trained over the same set of 300 trajectories.
5. There are quite a few typos in mathematical expressions. It makes difficult to verify the correctness of the theoretic results. Here are some examples.
5.1 In the soft value function of (5), \theta^i_{m,k+1} should be \theta^i_{m,k}.
5.2 In (9), \alpha should be time-varying.
5.3 In Theorem 1, \bar{\pi}^i_{(m,T)} and \bar{\theta}^i_{(m,T)} are never defined.
5.4 The notations in (18) and those of Step 1 on Page 8 do not match.
5.5 In Steps 1-3 on Page 8, the notations of the local soft value functions do not match.

**Questions:**

Please refer to the weaknesses.

---

> ### Author Response · Authors · 2024-11-30
>
> Thank you very much for your detailed feedback. Below, we address each identified weakness.
>
> **Weakness 1**: We acknowledge that our algorithm builds on the maximum likelihood IRL framework and employs FedAvg for the global updates. However, the problem we are solving is a bi-level optimization.  due to the reinforcement learning (RL) nature of the inner loop, federated learning requires careful adaptation to ensure compatibility with the RL algorithm. We believe that our contributions lie in the design of this tailored algorithm and the accompanying convergence analysis, which address these challenges in a meaningful way.
>
> **Weakness 2**: Thank you for suggesting additional papers on the topic. We are familiar with R1: Federated Inverse Reinforcement Learning for Smart ICUs with Differential Privacy, as it came up during our literature review. While R1 focuses primarily on application-specific aspects, it lacks a general problem formulation and does not provide theoretical guarantees. On the other hand, R2, R3, and R4 are imitation learning papers, which differ fundamentally from inverse reinforcement learning. R5 focuses on inverse constrained reinforcement learning, which adopts a different IRL formulation, and its solution could not be generalized to other IRL models such as ML-IRL, which we focused on. To the best of our knowledge, we are the first to study this specific problem for ML-IRL and provide theoretical convergence analysis.
>
> **Weakness 3**: For large-scale discrete state and action spaces, the Q-function and reward function can be represented by neural networks, and their parameters can be communicated across clients. In our approach, both the global policy and local policies are derived from the soft-Q function. As a result, there is no need to communicate policies directly; instead, we compute the global policy based on the aggregated Q-function, ensuring efficient and scalable communication.
>
> **Weakness 4**: Thank you for your observations regarding the experimental setup. In our experiments, we aim to demonstrate that federated learning provides a more effective solution compared to a purely local approach that is constrained by privacy concerns and limited to partial data. To reflect real-world dilemmas, we allow the federated framework to access diverse datasets across clients, whereas the centralized baseline is restricted to a single dataset. This setup is intended to highlight the advantages of federated learning in leveraging distributed data under privacy constraints.
>
> **Weakness 5**: Thank you for pointing out these issues. We address them as follows:
> 1. In Equation (5), $\theta^i_{m,k+1}$ should indeed be
>  $\theta^i_{m,k}$.
> 2. The time-varying property of $\alpha$ is mentioned in the proof to illustrate its role in determining the convergence rate.
> 3 & 4. In Theorem 1 Equation (18), $\bar{\pi}^i_{m,T}$ and $\bar{\theta}^i_{m,T}$ should be $\bar\pi_{m,T}$ and $\bar\theta_{m,T}$, respectively.
> 5. The typos in Steps 1-3 on Page 8 are corrected to make it consistent.
>
> Please check the updated supplement material for the updated full paper.
>
> We sincerely appreciate your comments and hope these clarifications address your concerns.

---

### Official Review · Reviewer_K1S3 · 2024-11-03

**Soundness:** 1
**Presentation:** 3
**Contribution:** 2
**Rating:** 3
**Confidence:** 4

**Summary:**

This paper proposes a federated learning framework for Inverse Reinforcement Learning in the discounted case, which performs IRL on each individual client and then aggregate the action-value functions and reward parameters.
There are numerous claims made in the paper appear to be invalid, which makes it questionable on the legitimacy of the proposed aggregation scheme and the theoretical claim, the majority of which itself is largely a repetition of existing analysis results.

**Strengths:**

The federal learning framework for Inverse Reinforcement Learning seems interesting. The proposed algorithm is intuitive and practical. The theoretical analysis is solid.

**Weaknesses:**

There are a few places the paper falls short.

> We note that when the $Q$-values are represented by another network with parameter $\psi$, the aggregation of the $Q$-values will simply become aggregation of model parameters.

1. The above claim made in line 282-283 implies that a neural network is linear. Correct me if I'm wrong but the common practice in federal learning is to aggregate the gradients of the functions, instead of the weights of the local updated model themselves.

2. The same claim is made for the reward function as well. Together with the first point, the proposed algorithm seems to have little to do with the theoretical results, except that if we consider a linear function approximation of both the $Q$-function and the reward estimation, which does not seems to be the focus of this work.

3. Since obtaining the average of the action-value function is trivialized in this paper, the proof of theorem 1 (lemma1 and 2 have already been shown in prior works, please cite properly) appear exactly the same for the most part (say, up until equation 51). Can authors elaborate the key difference and contribution of this work in analysis compared with the prior work of MLIRL?
4. Some experiments do not seem to make sense: in more than one case, a centralized agent presented with demonstrations of medium level of performance can learn a policy that performs the best.

**Questions:**

1. In Alg1, line 16-20, what is the index that's being looped over and how it affects the aggregation?
2. Can the authors say more on why standard federated learning framework can't work? Judging from the implementation, it seems that the proposed framework is also just a simple aggregation of objectives, which this case, the primal and dual objectives from IRL, i.e., the action-value function and reward function.
3. It seems that in the experiments section, the proposed algorithm has a large deviation of performance, while the table does not show the standard deviation. Is this intentional?

---

> ### Author Response · Authors · 2024-11-30
>
> Thank you very much for your thoughtful comments.
>
> **Weakness 1**: Thank you for your comment. For the proof, we focus on the tabular case where both the state and action spaces are finite, allowing Q-values to be aggregated. When extending to Q-Networks, both direct aggregation of the network parameters $\phi$ and aggregation of gradients are standard practices. For instance, the original FedAvg algorithm [1] aggregates neural network parameters, and existing work in federated reinforcement learning [2] also aggregates the parameters of both Q-Networks and policy networks.
>
> [1] McMahan, Brendan, et al. “Communication-efficient learning of deep networks from decentralized data.” Artificial Intelligence and Statistics. PMLR, 2017.
>
> [2] Jin, Hao, et al. “Federated reinforcement learning with environment heterogeneity.” International Conference on Artificial Intelligence and Statistics. PMLR, 2022.
>
> **Weakness 2**: Since the reward function is represented by a neural network, for similar reasons as weakness 1, we believe parameter aggregation is also valid for the reward function. Both the Q-function and the reward function, represented as neural networks, have sufficient capacity to generalize to functions of arbitrary complexity.
>
> **Weakness 3**: Thank you for looking into the details of our proof. The most challenging part of the convergence proof lies in addressing the influence of aggregation. Our proof focuses on the tabular case, similar to many existing work [3, 4] in this space aiming at optimality proofs. Prior ML-IRL work only considered a single agent problem and didn’t analyze it in a federated setting as how global aggregation impacts local models. On the other hand, traditional federated learning convergence analyses cannot be directly applied here, because the lower-level problem is a soft Q-learning RL problem and we do not have a gradient-based update rule for the tabular case. Therefore, our construction of convergence is based on soft Q-functions and the entropy-regularized Boltzmann policy updates, which prevents us from utilizing existing proof techniques in the federated learning area. Facing above challenges, we carefully conduct analysis to bound the errors caused by the aggregation of Q-values and reward parameters to evaluate the optimality of the inner-loop RL problem with respect to the Q-values and the outer-loop maximum likelihood estimation.
>
> [3] Mei, Jincheng, et al. "On the global convergence rates of softmax policy gradient methods." International conference on machine learning. PMLR, 2020.
>
> [4] Haarnoja, Tuomas, et al. "Soft actor-critic: Off-policy maximum entropy deep reinforcement learning with a stochastic actor." International conference on machine learning. PMLR, 2018.
>
> **Weakness 4**: In the experiment, we compared the centralized agent with medium-level performance to mixed expert demonstrations of varying performance levels in both federated and centralized settings. We found that demonstrations with lower performance can cause the recovered policy using mixed expert demonstrations to underperform compared to using only medium-level performance demonstrations. This phenomenon could be accounted for by the intuition that suboptimal demonstrations can adversely affect policy recovery.
>
> **Question 1**: In Algorithm 1, lines 16-20, the index being looped refers to the agent index.
>
> **Question 2**: Standard federated learning cannot be applied directly to inverse reinforcement learning (IRL). This has also been shown in several recent works that have developed decentralized IRL solutions [5, 6]. The inner RL problem typically has a unique form that is distinct from standard optimization problems. For instance, in our setup, the inner RL problem is formulated as soft Q-learning for Maximum Likelihood Inverse Reinforcement Learning. To ensure compatibility with this specific RL form, we decided to perform aggregation only on the Q-function rather than both the Q-function and the policy. This design choice simplifies the aggregation process and facilitates convergence.
>
> [5] Liu, Shicheng, and Minghui Zhu. "Distributed inverse constrained reinforcement learning for multi-agent systems." Advances in Neural Information Processing Systems 35 (2022): 33444-33456.
>
> [6] Reddy, Tummalapalli Sudhamsh, et al. "Inverse reinforcement learning for decentralized non-cooperative multiagent systems." 2012 ieee international conference on systems, man, and cybernetics (smc). IEEE, 2012.
>
> **Question 3**: The large deviations observed in our results may stem from differences between the environments. Due to the time and resource-intensive nature of the experiments, we were unable to run multiple trials. However, we plan to conduct further experiments and fine-tune the parameters to mitigate the effects of varying task settings. We appreciate your understanding and will incorporate this in future work.
>
> We sincerely appreciate your comments and hope these clarifications address your concerns.

---

### Meta-Review · Area_Chair_4spB · 2024-12-21

**Metareview:**

This paper studies Federated Maximum Likelihood Inverse Reinforcement Learning (F-ML-IRL) with a novel dual-aggregation and bi-level optimization technique to handle decentralized data. It provides a theoretical convergence analysis and empirical evaluation on MuJoCo robotic tasks. However, the reviewers found significant issues, including limited novelty, insufficient differentiation from prior works, poor presentation, and unaddressed critical comparisons to related literature. The rebuttal failed to convincingly address these concerns or demonstrate substantial technical contributions, leading to the conclusion that the paper does not meet the standard for acceptance.

**Additional Comments On Reviewer Discussion:**

During the rebuttal period, reviewers raised concerns about the paper's limited novelty, insufficient differentiation from prior work, incomplete experimental comparisons, poor presentation, and lack of clarity on unique challenges and contributions. Reviewer AXzk questioned the theoretical novelty and demanded clearer articulation of unique technical challenges, while Reviewer QuNC criticized the presentation quality and lack of technical depth. Reviewer K1S3 highlighted inconsistencies in the theoretical claims and insufficient empirical validation, and Reviewer usqH requested comparisons to existing distributed IRL methods and better experimental justification. The authors attempted to address these points by elaborating on the convergence proof, clarifying presentation issues, and defending the relevance of their contributions in the context of ML-IRL. However, these responses were largely deemed insufficient, as key concerns about novelty, rigor, and adequate experimental evidence remained unresolved.

---

### Decision · Program_Chairs · 2025-01-22

Reject